# Chronic platelet-derived growth factor receptor signaling exerts control over initiation of protein translation in glioma

Shuang Zhou[1], Vicky A Appleman[1], Christopher M Rose[6], Hyun Jung Jun[1], Juechen Yang[3], Yue Zhou[4], Roderick T Bronson[5], Steve P Gygi[6], Al Charest[1,2]

**Activation of the platelet-derived growth factor receptors (PDGFRs) gives rise to some of the most important signaling pathways that regulate mammalian cellular growth, survival, proliferation, and differentiation and their misregulation is common in a variety of diseases. Herein, we present a comprehensive and detailed map of PDGFR signaling pathways assembled from literature and integrate this map in a bioinformatics protocol designed to extract meaningful information from large-scale quantitative proteomics mass spectrometry data. We demonstrate the usefulness of this approach using a new genetically engineered mouse model of PDGFRα-driven glioma. We discovered that acute PDGFRα stimulation differs considerably from chronic receptor activation in the regulation of protein translation initiation. Transient stimulation activates several key components of the translation initiation machinery, whereas the clinically relevant chronic activity of PDGFRα is associated with a significant shutdown of translational members. Our work defines a step-by-step approach to extract biologically relevant insights from global unbiased phospho-protein datasets to uncover targets for therapeutic assessment.**

## Introduction

The last few decades have witnessed intense efforts in deciphering molecular events that contribute to the transmission of extracellular signals to intracellular physiological responses. Intricate networks of positive and negative regulatory mechanisms transmit these signals, and this immense complexity impacts our ability to interpret and predict cellular outcomes of specific inputs. Overcoming this knowledge gap is essential to better understand diseases and apply new therapeutic approaches. A crucial requisite

toward unraveling this complexity is the use of network analysis, computational modeling, and visualization tools. At the apex of this endeavor stands the assembly of networks constructed from manually curated information garnered from literature and translated into computer-readable databases (such as Biological PAthway eXchange [BioPAX; www.biopax.org] and Systems Biology Markup Language [SBML; http://sbml.org/] [Hucka et al, 2003]). This approach has been used to create detailed signaling maps for epidermal growth factor (EGF) receptor (EGFR) (Oda et al, 2005), mechanistic target of rapamycin (mTOR) (Caron et al, 2010), Rb/E2F pathway (Calzone et al, 2008), toll-like receptor signaling (Oda & Kitano, 2006), and comprehensive molecular interaction maps for budding yeast cell cycle (Kaizu et al, 2010) and rheumatoid arthritis (Wu et al, 2010).

The platelet-derived growth factors (PDGFs) are a family of growth factors (PDGF-A, -B, -C, and -D) that controls the growth of connective tissue cells and are critical regulators of mesenchymal cells during embryonic development (reviewed in Kazlauskas [2017]). Homo- or heterodimers of PDGF ligands activate two types of cell surface receptor tyrosine kinases (PDGFRα and PDGFRβ) by inducing homo- and heterotypic dimerization (αα, αβ, or ββ) to elicit various intracellular signaling pathways and physiological responses (reviewed in Chen et al [2013]). Deregulation of PDGF–PDGFR signaling leads to a number of diseases, including many types of cancers (Ostman, 2004). Although several signaling pathways downstream of acutely activated PDGFRs are known, this knowledge remains relatively rudimentary, which hampers the development and full understanding of PDGFR signaling networks. A comprehensive agglomerated map of all known PDGFR signal transduction pathways is nonexistent and would represent a valuable resource to the research community.

The genomic landscape of glioblastoma multiforme (GBM) revealed a number of genetic mutations and signaling abnormalities that are known drivers of cancer (Cancer Genome Atlas Research Network, 2008; Verhaak et al, 2010; Brennan et al, 2013). Amplification, overexpression, and mutations of *EGFR* and *PDGFRA* are among the most

[1]Cancer Research Institute, Beth Israel Deaconess Medical Center, Boston, MA, USA    [2]Department of Medicine, Harvard Medical School, Boston, MA, USA    [3]Department of Computer Science, North Dakota State University, Fargo, ND, USA    [4]Department of Statistics, North Dakota State University, Fargo, ND, USA    [5]Rodent Histopathology Core, Dana-Farber/Harvard Cancer Center, Boston, MA, USA    [6]Department of Cell Biology, Harvard Medical School, Boston, MA, USA

Correspondence: acharest@bidmc.harvard.edu

common genetic aberrations of receptor tyrosine kinases in GBM occurring in 57.4% and 13.1% of patients, respectively (Brennan et al, 2013). These oncogenic drivers are paired with characteristic homozygous deletion or mutation in the tumor suppressor genes INK4a/ARF (*CDKN2a*) and p53 and loss of PTEN function in most cases. Recapitulating these genetic events in mice has generated several accurate models of glioma that have proven to be instrumental in advancing new concepts in gliomagenesis (reviewed in Hambardzumyan et al [2011]). The current standard of care for GBM patients does not include driver mutation–based precision medicine interventions because they have been unsuccessful in treating GBM, likely owing to redundancy in signaling pathways (reviewed in Olson et al [2014] and Prados et al [2015]). There is a clear need for a deeper understanding of signaling pathways in GBM, which we have previously achieved using mouse models where driver signaling events (e.g., EGFR) are fully controlled (Zhu et al, 2009; Acquaviva et al, 2011; Jun et al, 2012). Similarly, progress in PDGFR signaling in glioma would greatly benefit from an in vivo model that is based on clinically relevant parameters such as overexpression of PDGFRα and its chronic stimulation by PDGF-A ligand.

Our knowledge of signaling pathways in general is heavily biased toward signal components that (i) have historically been the focus of attention (e.g., MAPK and PI3K) and (ii) are associated with the availability of specific reagents (e.g., phospho-specific antibodies). These biases are typically overcome by the use of more global approaches such as tandem mass spectrometry (MS) phospho-proteomic techniques (Dephoure et al, 2013); however, there remain significant limitations for many interesting MS-generated phospho-events because of unassigned biological functions for most phosphosites. This lack of annotated functionality poses a significant problem to the use of MS-based generation of signaling networks, and well-delineated processes to extract practical information from phospho-databases are under development (Terfve et al, 2015; Munk et al, 2016a, b; Refsgaard et al, 2016).

Here, we depict a comprehensive signaling map of PDGFR signaling and describe a bioinformatics strategy to extract biologically relevant information from large phospho-proteomics datasets. We applied our approach to a new genetically engineered mouse model of PDGFRα-driven GBM, and we discovered significant changes in the activity of translational initiation machinery signaling between acutely versus chronically stimulated PDGFRα GBM, which can have therapeutic implications.

# Results

### Activity flow and process diagrams of PDGFR signaling

Visual representations of signaling pathways tend to be fragmented, incomplete, and often simplistic. To aid visualizing all known PDGFR signaling events, we constructed a series of comprehensive maps of PDGFRα and β signal transduction pathways by manually curating published literature (details of the curation process can be found in the Supplementary Information). The PDGFR networks are assembled in human- and computer-readable formats using the CellDesigner software platform (http://celldesigner.org/) (Kitano et al, 2005), which offer well-defined and coherent graphical annotations to depict signaling interactions. In addition, CellDesigner allows users to access the references that were used to source the information for each individual reaction using PubMed ID (PMID). Furthermore, the data describing the molecular interactions are stored in and comply with SBML and the Systems Biology Graphical Notation (SBGN) process diagram (Kitano et al, 2005; Le Novere et al, 2009) for machine readability and graphical representation, respectively.

We first created a global activity flow map representing all four PDGF ligands and their interactions with all three receptor combinations (αα, ββ, and αβ dimers) and the signaling pathways that they activate (Fig 1). This map is similar to common annotations used in the current literature, and it simply depicts the flow of information between biochemical species in the signaling network. It shows the affinity of homo- and heterodimer ligand combinations for receptor dimers and their known first-level signaling partners. The activation of immediate early signaling members such as Grb2/7, CRK, SHP2, PI3K, PLCγ, Syp, SRC, SHC, NCK, RasGAP, VAV2, and STAT1/3/5, in turn leads to the establishment of signaling networks that ultimately result in changes in cellular physiology mediated by changes in transcription of target genes, translation biology, invasion and actin cytoskeleton organization, autophagy, mitogenesis/cell cycle, endocytosis and degradation/recycling of PDGF receptors, and angiogenesis (Fig 1).

This general activity flow map is useful for a broad overview of signaling. However, it lacks information about the state transitions of signaling entities. Therefore, we expanded the activity flow representation into two separate maps that include state transition details for all signaling components (Figs 2 and S1). Given the nature of PDGFRαα and ββ homodimerization, we constructed maps that separately represent each dimer. To assist in exploring the maps, we organized them in distinctive functional modules, including regulation of PDGFR endocytosis, degradation or recycling, calcium signaling, cap-dependent translation, cytoskeleton dynamics, transcription and cell cycle, small GTPase-mediated signal transduction, MAPK cascade, phosphatidylinositol polyphosphate signaling, and mTOR signaling (Figs 2 and S1). In these process diagrams of PDGFR, the phosphorylation states of all signaling members are represented in a system of active or inactive transitive nodes.

Collectively, the maps comprise 648 species and 481 reactions. In SBML, a species is defined as an entity that is the subject of a reaction. It is used to represent the various states that are the result of enzymatic modifications, protein complexes association/dissociation, and translocation. The maps were manually constructed based on 390 publications that are available in the Supplementary Information, and for individual reactions, PMID are added, enabling a direct link to the relevant references. The guidelines for the curation process are described in the Supplementary Information. The information of the components in these maps has been summarized in Table S1. In terms of phosphorylation events, combined, these maps contain 189 annotated phosphosites (149 Ser/Thr and 40 Tyr) on 56 proteins. The reactions were also matched with the Reactome database (https://reactome.org/), and the coverage between our map and the Reactome database is summarized in Table S2. We designed the maps to be self-explanatory and they represent a comprehensive view of PDGFR

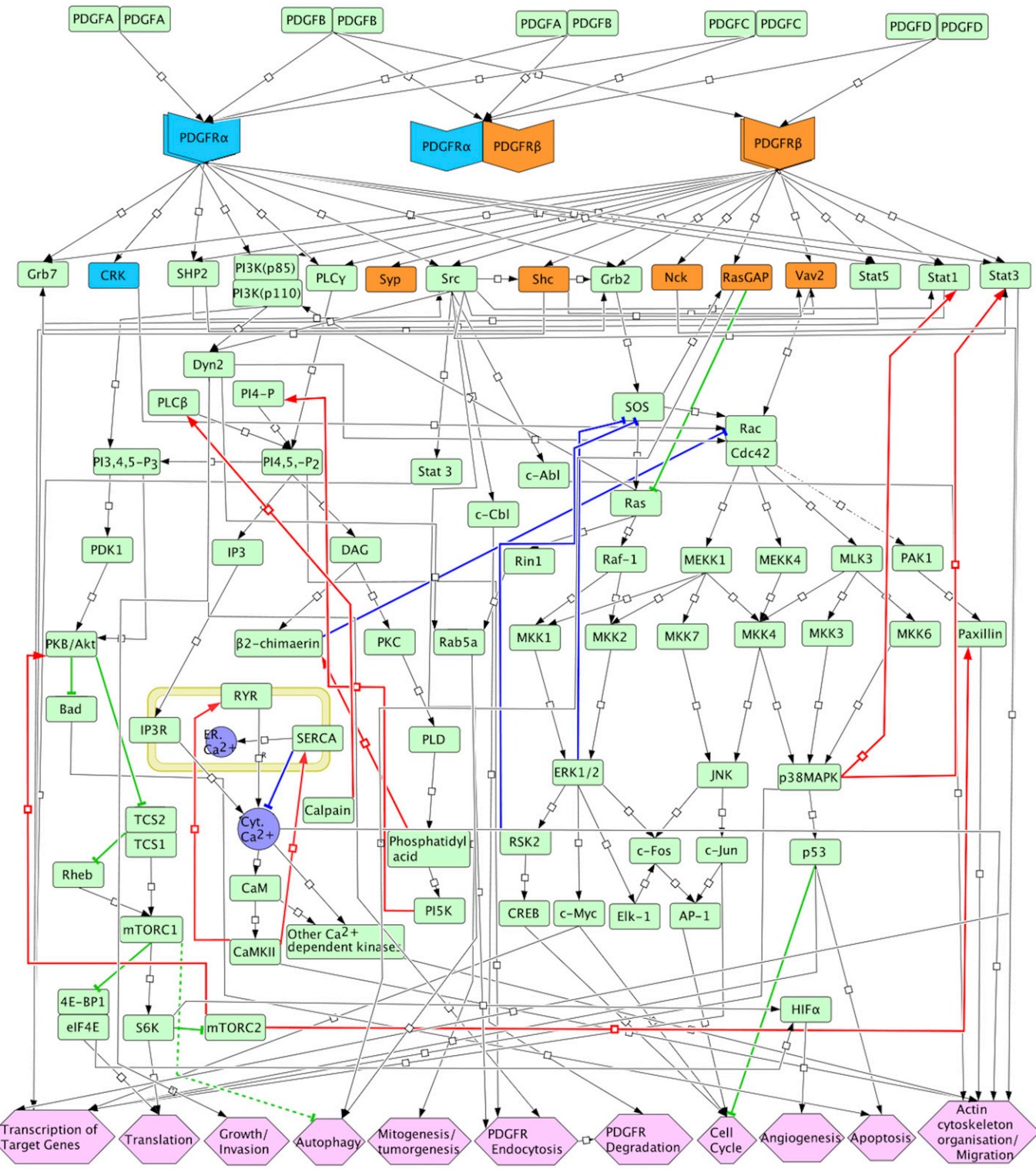

**Figure 1.  Activity flow of PDGFR signaling pathways.**
Activity flow diagram of broad PDGFR signaling. Components and reactions that are central to PDGFR signaling were extracted from the more comprehensive maps (Figs 2 and S1) to produce a reduced complexity map highlighting the flow of activation and inhibition of signaling members. The resulting downstream activities of the core components have significant roles in controlling diverse biological responses. Arrows in the diagram represent the flow of reaction with positive (red) and negative feedback (blue) loops represented by bold lines. Direct downstream molecules uniquely regulated by PDGFRα and PDGFRβ receptors are indicated in blue and orange, respectively. Symbols are identical to those used in Figs 2 and S1 legends. The high-resolution PDF is available in the Supplementary Information.

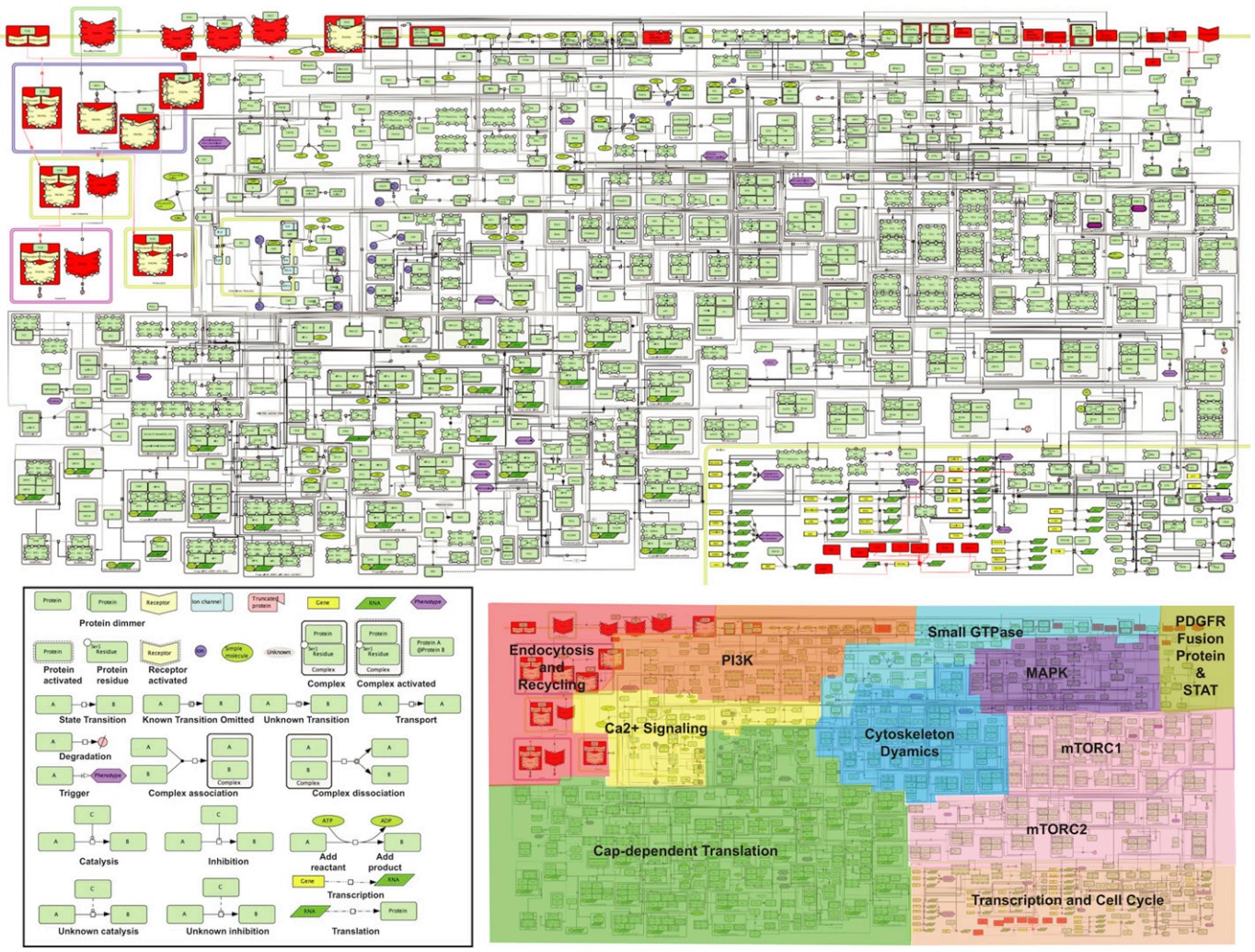

**Figure 2. A comprehensive molecular interaction map of PDGFRαα signaling network.**
Detailed graphical representation of PDGFRαα signaling events. The map was created with CellDesigner version 4.4 (http://www.celldesigner.org). Included are a total number of 615 species and 448 reactions extracted from 390 publications (Supplementary Information). PMIDs are included for individual reactions, which enables a direct link to the relevant literature. The information for all the species, compartments, and reactions can be viewed using the SBML version of the map. The symbols adopted to build the map are illustrated in the legend shown in the left bottom. Red-colored items are unique molecules present in PDGFRαα network only and absent from the PDGFRββ comprehensive map (Fig S1). The SBML and PDF files of the PDGFRαα network are available in the Supplementary Information.

signaling. Signaling pathways are highly context dependent, and a limitation to these maps is the absence of contextual information. To further improve PDGFR signaling networks in a clinically relevant context, we created a mouse model of glioma based on over-expression and constitutive activation of PDGFRα.

## Autocrine/paracrine constitutive activation of PDGFRα in mice generates GBM

Approximately 13% of adult GBMs overexpress PDGFRα (Brennan et al, 2013). To study PDGFR signaling in this clinical context, we created a mouse model of GBM based on overexpression and chronic activation of PDGFRα. We generated a conditional transgenic mouse strain (LoxSTOPLox-PDGFRα) capable of overexpressing the human PDGFRα receptor in a Cre recombinase–dependent manner (Fig 3A). In GBM, PDGFR signaling is chronic because of autocrine/paracrine

ligand stimulation (Hermansson et al, 1988; Maxwell et al, 1990; Hermanson et al, 1992; Vassbotn et al, 1994; Guha et al, 1995). To mimic this clinical state, we used a lentivirus that delivers a doxycycline (DOX)-inducible expression of human PDGF-A ligand along with a constitutive expression of Cre (Fig 3A). In patients, PDGFRα-positive GBMs are often observed within the context of loss-of-function TP53 mutations (Cancer Genome Atlas Research Network, 2008; Verhaak et al, 2010; Brennan et al, 2013). We, therefore, bred our LSL-PDGFRα strain to a conditional p53$^{2lox}$ strain (Marino et al, 2000) to better reflect the genetics of driver mutations observed in patient GBMs.

Brain tumors were created by stereotactic intracranial injections of the aforementioned lenti PDGF-A-Cre viruses in adult (>12 wk) LSL-PDGFRα;p53$^{2lox}$ mice that were fed a DOX diet (250 mg/kg) ad libitum post-injection (Fig 3B–D). We found that lethal tumors developed in a PDGF-A ligand-dependent manner (Fig 3C). Every LSL-PDGFRα;p53$^{2lox}$ animal injected with lenti PDGF-A-Cre virus fed

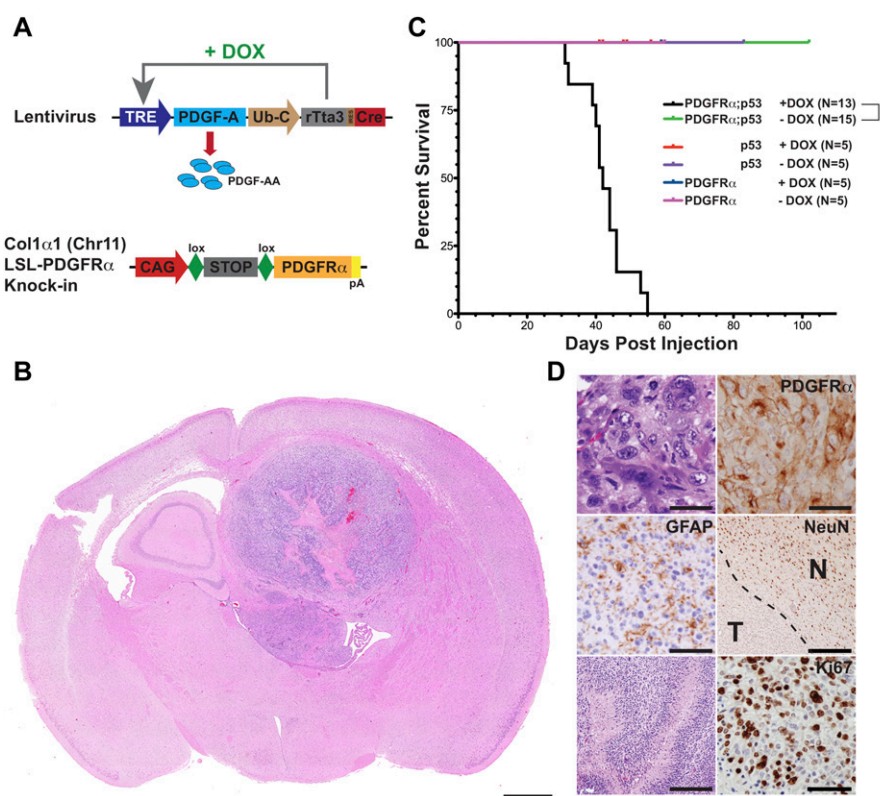

**Figure 3. Conditional expression and activation of PDGFRα in the brain of adult mice produces glioblastoma.**
**(A)** Schematic representation of the Cre–Lox conditional PDGFRα transgene knocked-in the 3′ UTR of the Col1α1 locus. The transcriptional activity of the CAG promoter is prohibited by the presence of a floxed stop cassette (LSL). Expression of PDGFRα is achieved by intracranial stereotactic injections of a Cre recombinase–expressing lentivirus. The virus also contains a DOX-inducible PDGF-A ligand. DOX administration induces expression of PDGF-A and chronic receptor kinase activation. **(B)** Representative photomicrograph of an H&E-stained formalin-fixed paraformaldehyde embedded section of a PDGF-A; PDGFRα;p53$^{-/-}$ brain tumor. Scale bar, 1 mm. **(C)** Tumor-free survival (Kaplan–Meier) analysis of cohorts of mice of the indicated genotypes fed a DOX diet of 250 mg/kg. *P < 0.0001 log-rank (Mantel–Cox) test. **(D)** PDGF-A; PDGFRα;p53$^{-/-}$ tumors have histopathological features of glioblastoma. Representative photomicrographs of formalin-fixed paraformaldehyde embedded tumor sections stained with H&E and immunohistochemical detection of PDGFRα, glial fibrillary acidic protein (GFAP), NeuN, and Ki-67. Scale bars: top row—H&E and PDGFRα 62.5 μm; middle row—GFAP 125 μm, and NeuN 250 μm; and bottom row—H&E and Ki67 62.5 μm. T, tumor; N, normal brain; GFAP, glial fibrillary acidic protein.

a DOX diet succumbed from intracranial tumors (Fig 3C) with a median survival of 42 d, whereas none of p53$^{2lox}$ or LSL-PDGFRα control animals did (Fig 3C). We also observed that simple overexpression of PDGFRα in the context of p53 nullizygocity is insufficient to generate intracranial tumors (no DOX controls), demonstrating an absolute requirement for kinase activity of PDGFRα for tumorigenesis. In addition, none of the PDGF-A-Cre–injected p53$^{2lox}$ and LSL-PDGFRα no DOX control mice developed tumors after >80 d post-injection, at which point the experiment was terminated and necropsy performed to validate the absence of tumor (data not shown). The PDGF-A;PDGFRα;p53$^{-/-}$ tumors displayed histopathological features that are observed in high-grade gliomas, i.e., tumors were densely packed, containing poorly differentiated cells with marked nuclear atypia, numerous giant multinucleated cells, and characteristic areas of pseudopalisading necrosis (Fig 3D). Cells within these high-grade glioma tumors expressed the astrocytic marker glial fibrillary acidic protein, lacked expression of the neuronal marker NeuN, and displayed high level of the proliferation marker Ki67 (Fig 3D). These histopathological features are consistent with those observed in GBM. This bipartite system provides a strict, spatiotemporal control over PDGFRα expression and chronic activation to study PDGFRα-driven signaling events during GBM tumorigenesis.

## Unique PDGFRα signaling properties under chronic and acute stimulation

Most, if not all, of our knowledge of PDGFR signaling pathways has been generated using ex vivo culture systems, whereupon receptor kinase activity is triggered using acute, saturating levels of ligand stimulation. This approach has generated very informative data on signaling pathways; however, it does not fully mimic the clinical setting, in which PDGFR signaling is a chronic, autocrine/paracrine type of stimulation. To address this matter, we studied acute and chronic PDGFRα stimulation in primary cell cultures derived from the PDGF-A;PDGFRα;p53$^{-/-}$ GBM tumors. These cultures demonstrate a titratable and temporal DOX-dependent expression of PDGF-A (Fig S2A and B). In addition, they retain overexpression of human PDGFRα (Fig S2C and D) and chronic expression of DOX-stimulated PDGF-A ligand result in activation of the receptor kinase activity as measured by the levels of PDGFRα autophosphorylation site Tyr849 (Fig S2C and D). In the absence of DOX, acute stimulation of these cultures by addition of exogenous recombinant human PDGF-AA results in a robust activation of PDGFRα kinase activity (Fig S2C and D). These cultures enable the study of chronic and acute signaling pathways in the same cellular context and allow for a direct signaling comparison between the two types of input stimulation. We determined that both acute treatment (15 min with 25 ng/ml) with exogenous recombinant PDGF-AA and chronic exposure (48 h with 10 μg/ml DOX) to PDGF-AA resulted in similar activation levels of PDGFRα kinase activity (Fig S2C and D).

We gained insight into signaling events initiated by acute and chronic receptor stimulation by performing an unbiased and global comprehensive quantitative proteomic and phospho-proteomic (pTyr, pSer, and pThr) analysis under acute (15 min, 25 ng/ml PDGF-AA) and chronic (48 h, 10 μg/ml DOX) PDGF-A stimulation in comparison with control, unstimulated cells. We performed isobaric

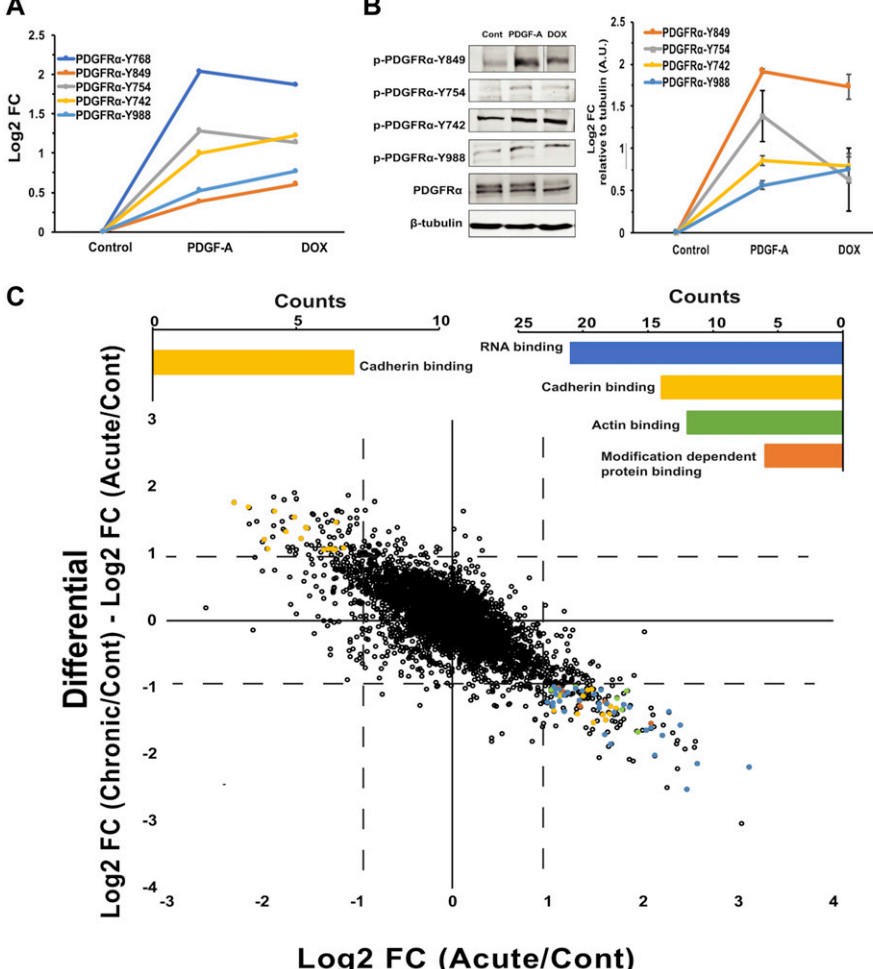

**Figure 4. Phospho-proteomic analysis of acutely and chronically stimulated PDGFRα signaling.**
**(A)** Log$_2$ fold change (FC) of PDGFRα autophosphorylation in MS data. **(B)** Quantitative Western blot analysis for PDGFRα autophosphorylation. **(C)** Analysis for differential log$_2$FC on phosphorylation upon acute or chronic stimulation of PDGFRα. On the left and right top are the potential most differentially up- (left) and down-regulated (right) signaling functionalities upon chronic versus acute stimulation of PDGFRα identified by GO term enrichment analysis. Only the categories with *P*-value < 0.05 are displayed. False discovery rate (FDR) < 0.05. Source data are available for this figure.

label-based quantitative MS to measure expression of proteins and phospho-isoforms (Ting et al, 2011; McAlister et al, 2014) on pooled biological replicates (N = 3). This yielded expression data for 7,528 proteins across all conditions. To measure the alterations of signaling pathways between acute and chronic receptor stimulation, we performed successive phosphopeptide enrichment and pTyr immunopurification to quantify 5,767 phosphopeptides, of which 195 contained a tyrosine phosphorylation event. The phosphosites were found on 1,847 unique proteins and for the majority of these proteins (73.6%), 1–3 phosphorylation sites were detected per protein, with the most number of phosphorylation sites on a single protein being 106 (protein Srrm2) (Fig S3A). Proteins with the fewest number of phosphosites detected appear to be smaller (composed of fewer amino acids) than those with many more phosphosites (Fig S3A).

Because protein expression can influence measurements of enriched posttranslational modifications (PTMs), we normalized quantitative values for phosphorylation sites to protein-level expression measured in un-enriched samples and calculated differences between stimulated (acute and chronic) and control as log$_2$ fold change (FC) for each phosphopeptide. We first compared the acutely versus chronically stimulated PDGFRα autophosphorylation sites Y742, Y754, Y768, Y849, and Y988. Acute stimulation of

PDGFRα results in increased levels (log$_2$FC when compared with control unstimulated) on all sites, and chronic stimulation of the receptor led to similar levels of sustained phosphorylation for all sites (Fig 4A). We validated these results using phospho-specific PDGFRα antibodies by quantitative Western blot analysis and showed that, with the exception of pY754, the levels of phosphorylation for these sites among acute and chronic receptor activity were similar when compared to control unstimulated cells (Fig 4B). We extended this observation to additional, independent primary cell cultures derived from PDGFRα;p53$^{-/-}$ GBM tumors (Fig S3B–F) and demonstrated the universality of these observations.

We investigated the differences between chronic versus acute activation of PDGFRα signaling in a more global fashion. We analyzed the log$_2$FC of acute stimulation phosphorylation events versus control unstimulated. 50.3% of phospho-events showed an increase (log$_2$FC > 0) in levels and 49.7% displayed a decrease in levels after 15 min of PDGF-A stimulation (Fig 4C, *x*-axis). We then compared acute with chronic PDGFRα stimulation by analyzing the differential between chronically stimulated and acutely stimulated samples (Fig 4C, *y*-axis). The majority (96.5%) of the phosphopeptides display minimal changes (log$_2$FC between −1 and 1) between control, acute, and chronic ligand stimulation; however, 3.5% of the

phosphopeptides showed variations (log2FC > +1 or < −1) between chronic and acute PDGF-A stimulation (Fig 4C). For the 3.5% of most differentially regulated phospho-events, phosphosites that increased in intensity upon acute receptor stimulation show considerable decrease in intensity upon chronic receptor activation, and similarly, phosphosites that were decreased upon acute receptor activation displayed increases in intensity upon chronic stimulation. Events displaying the most drastic changes are listed in Table S3. This suggests that constitutively activated PDGFRα emits steady-state signaling that is significantly different than bolus, saturating short-term receptor activation. To gain meaningful insights into functionalities for these differences, we performed Gene Ontology (GO) term enrichment analysis on proteins from outlier phosphorylation events ($log_2FC < −1, > +1$) (Fig 4C, insets). Of the top-ranking categories are genes coding for proteins that are associated with regulation of protein translation and cadherin signaling. Phosphopeptides from these proteins were inversely correlated with the mode (acute versus chronic) of PDGFRα activity.

## Chronic PDGFRα activation orchestrates repression of protein translation initiation

Unbiased phospho-MS–based approaches typically generate informative data; however, functionality for most of the MS-derived phosphosites is unknown. Therefore, to gain mechanistic insights into the signaling differences between acute and chronic PDGFRα activation, we developed a method to systematically extract, from any given phospho-databases, events that have known, assigned functions. As a starting point, we used the PhosphoSitePlus public database (Hornbeck et al, 2015), which is a comprehensive, systematically curated resource of phosphorylation data from human, mouse, and rat proteins (https://www.phosphosite.org). We integrated our PDGFRα phospho-MS data with datasets in PhosphoSitePlus using an in-house algorithm coded in python programming language (Fig S4A and the Supplementary Materials and Methods). At the time of writing, PhosphoSitePlus comprised 326,502 phosphosites on 22,631 independent proteins and a dataset of 10,812 kinase-annotated phosphosites on 2,978 proteins that are the substrates of 408 kinases (Fig S4B). Note that the majority (96.7%) of the 326,502 phosphosites are functionally unannotated events, meaning the identity of the kinase(s) responsible for their phosphorylation is undetermined. Of the 540 kinases in the mouse genome, 408 are represented in the database (Fig S4C). Cross-referencing the phosphosites extracted from the Celldesigner PDGFR signaling maps (Figs 2 and S1) to the PhosphoSitePlus dataset reveals 95% concordance (180/189 phosphosites). This leaves nine unannotated phosphosites on six proteins that are not found within the PhosphoSitePlus database. By comparing our MS phosphosites dataset with the PhosphoSitePlus dataset, we determined that 90.3% (5,209/5,767) of the sites were matched within the 326,502 dataset, leaving 558 phospho-events on 283 proteins (Fig S5A), which represent newly identified phosphorylation sites.

To functionally annotate our PDGFRα MS data, we merged our MS-derived 5,767 phosphosites to the kinase substrate dataset of PhosphoSitePlus. This revealed that kinase–substrate information is available for 346 phosphosites on 241 proteins and that 150

kinases are associated with these 346 phosphosites (Fig S5B). The majority of these proteins (174/241, 72.2%) contain only one kinase-annotated phosphorylation site, and the largest number of phosphorylation sites on a single protein was nine (protein Ahnak) (Fig S5C). To uncover significant differences in phospho-events, we first classified the phosphosites/proteins into different subgroups based on our PDGFRα map and according to their molecular functions. We then performed a Kolmogorov–Smirnov test among the different functional groups and kinase–substrate groups to establish significance. Among all the functional groups, RNA binding pathway showed the most significantly different distribution under acute versus chronic PDGFRα stimulation ($P < 0.05$) (Table S4). Within this group, the most differentially influenced kinases included mTOR, p90RSK, ERK2, and JNK3 (Table S5).

Finally, to explore phospho-differences between chronic and acute activation of PDGFRα signaling, we used the Cytoscape visualization software designed to integrate biomolecular interaction networks in a unified conceptual framework (Shannon et al, 2003). We classified the phosphosites/proteins into different subgroups based on our PDGFRα map and according to their molecular functions. The strength of the connection between these phosphorylation sites and their matching kinases was set using a scale derived from the quantitative $log_2FC$ values (treated versus control untreated) from the MS data. The network visualization clearly shows a different pattern for chronic versus acute activation of PDGFRα signaling, with the "RNA-binding and translation-related" signaling being the most differentially regulated signaling pathway (Fig 5A).

Whereas acute activation of PDGFRα increased the phosphorylation levels of Rps6, Pdcd4, Eif4b, and Rbm14, chronic stimulation led to significant decreases in these same phosphosites (Fig 5B). We validated these observations using quantitative Western blot analysis (Figs 6A and S6). In concert with the MS data (Fig 6B), the phosphorylation of eIF4B S422, PDCD4 S457, S6 ribosome protein S235/236/240, and 4EBP1 S64 was increased or sustained in acute PDGFRα activation signaling when compared with control but was significantly reduced under chronic stimulation conditions (Fig 6C). This suggests a differential regulation of protein translation under conditions of acute and chronic PDGFRα activation. p90RSK and p70S6K are kinases known to phosphorylate eIF4B, PDCD4, and S6 ribosomal proteins. We analyzed the phosphorylation levels (a surrogate of activation status) of these upstream kinases by Western blot and showed that both were considerably diminished in chronic activation of PDGFRα cells compared with acute activation (Fig 6D and E). These results suggest that constitutively active PDGFRα attenuates its signaling toward translational repression. Finally, we interrogated the the cancer genome atlas GBM reverse phase protein array database and found that high PDGF-A–expressing GBMs have statistically significant lower levels of phospho-S6K T389 ($P = 0.0006$) and trending lower levels for phospho-4EBP1 S65, RPS6 S235/236, S240/244, and p90RSK T359/363 (Fig 6F). These results demonstrate that GBM patients with chronic PDGF-A expression have lower phosphorylation levels of several key components of the translational machinery that collectively would result in decreased translational capacity.

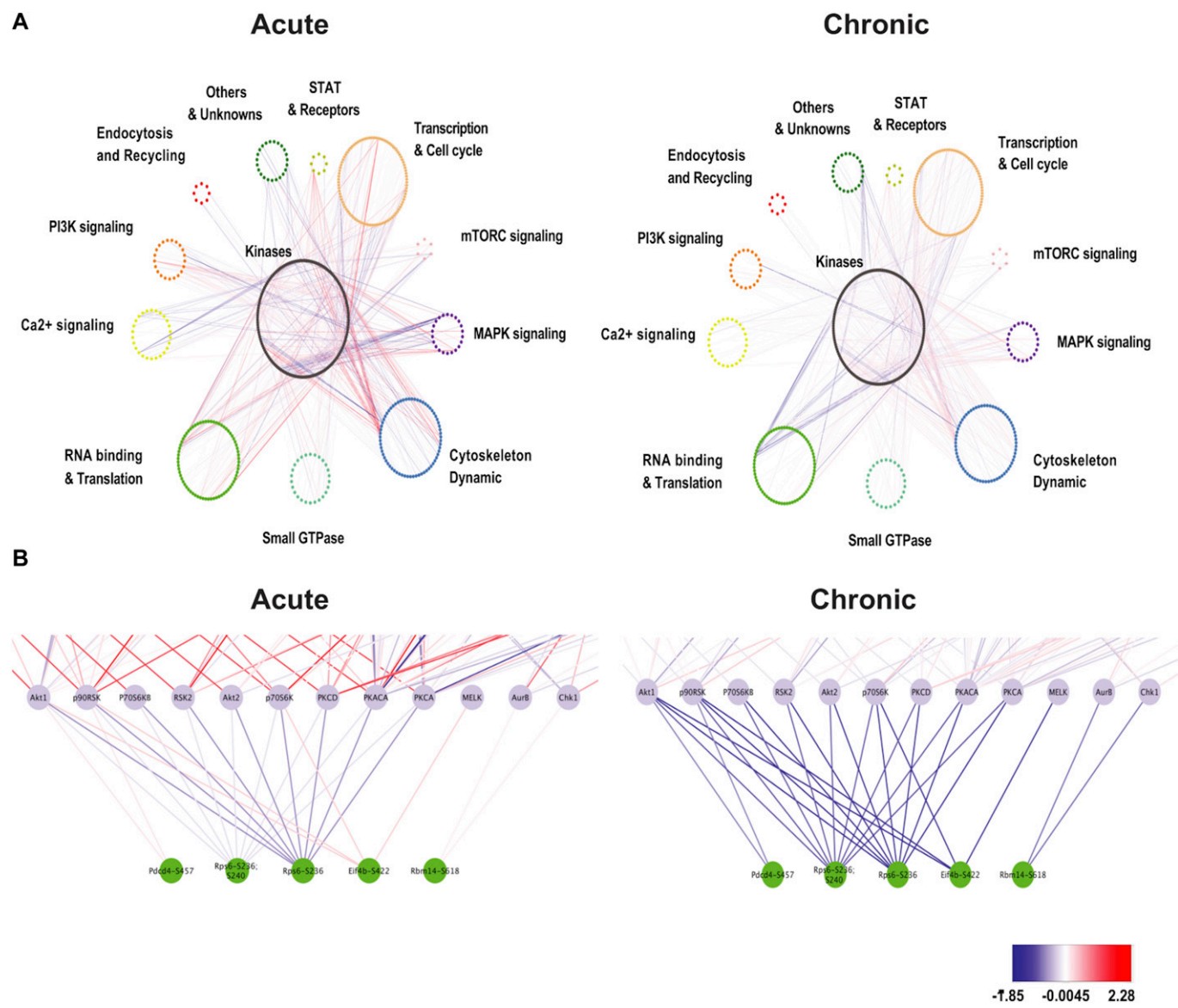

**Figure 5. Visualization of differential changes between chronic versus acute PDGFRα stimulation.**
**(A)** Cytoscape output of kinase–substrate network visualization of acute and chronic stimulation of PDGFRα. Source data were obtained by extracting kinase–substrate pairs from the PDGFRα MS data based on the Kinase_Substrate_Dataset. Phosphorylation sites were classified according to their molecular functions and assigned color codes that are consistent with those in Fig 2. The color of the edges represents the quantitative log$_2$FC values from the MS data of stimulated to control unstimulated. **(B)** Close-up view of the most inhibited molecules and corresponding kinases in the category of "RNA binding and translation." The Cytoscape format file of the network visualization is available in the Supplementary Information.

## Chronic PDGFRα activation reduces sensitivity of PDGFRα-positive GBM cells to translational inhibitor–induced suppression of cell viability

Deregulation of mRNA translation is a common feature of cancer and represents an emerging class of cancer drug targets. Several inhibitors targeting the mRNA translation regulation have demonstrated anticancer activities in preclinical and clinical cancer models (Bhat et al, 2015). To further understand the biological significance of chronic PDGFRα activation in GBM, we investigated the effects of chronic PDGFRα activation on drug response to translational inhibitors. Three translation inhibitors, 4EGI-1 (Moerke

et al, 2007), LY2584702 (Tolcher et al, 2014), and AZD8055 (Chresta et al, 2010), which target the translation initiation eif4F complex, the p70S6K, and mTORC, respectively (Fig S7), were applied to chronically activated PDGFRα GBM primary cell cultures and cell viability was measured. We found that chronic PDGFRα activation (48 h, 10 μg/ml DOX) confers resistance to translational inhibitor–induced suppression of cell viability (Fig 7A–C). To control for off target effects of DOX, we treated EGFR GBM primary cell cultures (Zhu et al, 2009; Acquaviva et al, 2011; Jun et al, 2012) with DOX (10 μg/ml) for 48 h and showed no significant impact on the cell viability (Fig 7D–F). This confirms that the resistance to translational inhibitor is dependent on chronic PDGFRα activation. These results

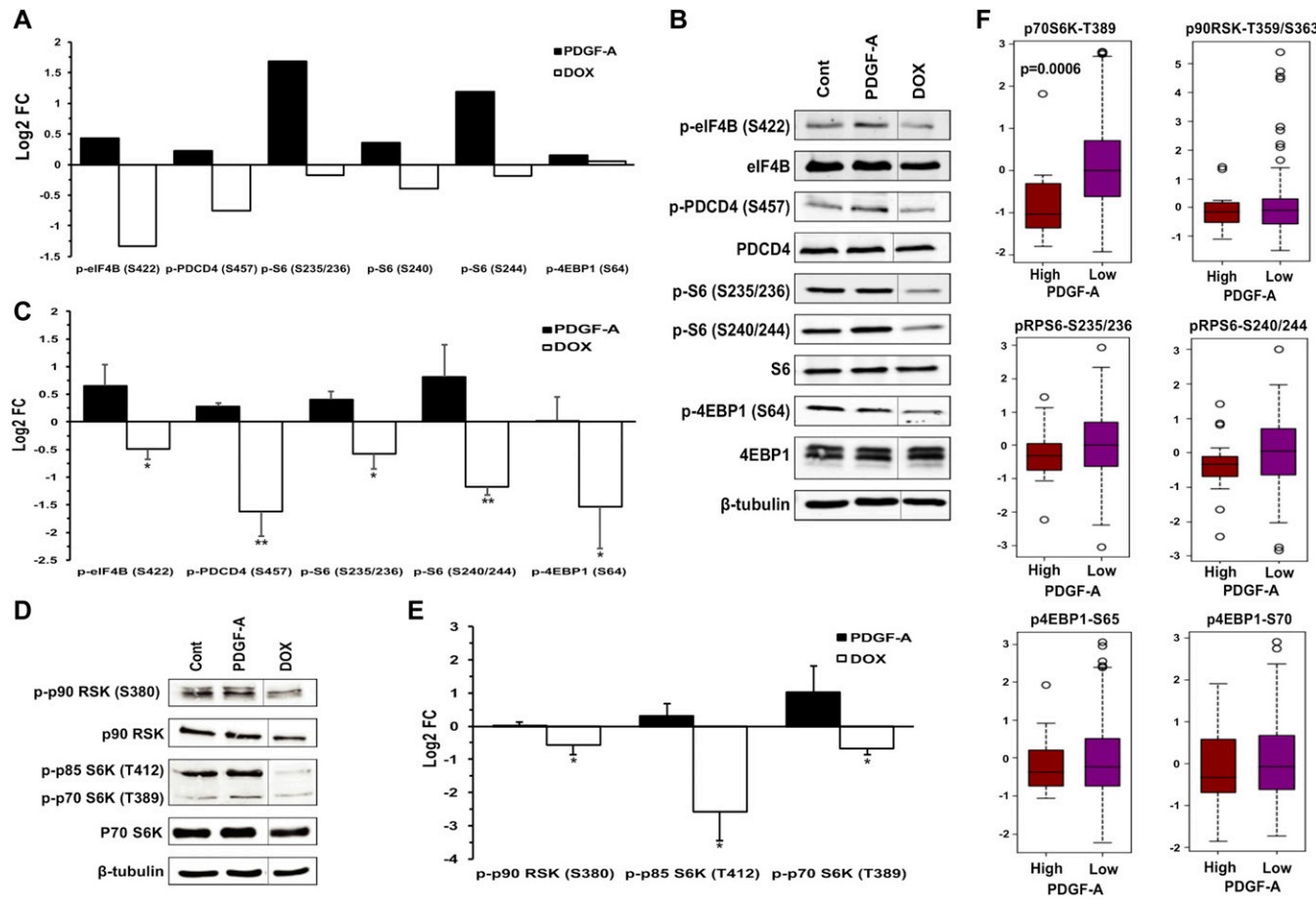

**Figure 6. PDGFRα activity reduces phosphorylation of critical members of the translational initiation complex.**
**(A)** Log$_2$FC of the key members of the translational initiation complexes in acute and chronic PDGFRα stimulation. **(B)** Western blots of the phosphorylation sites shown in (A) from biological replicates (N = 3). **(C)** Quantification of the Western blotting (B). **(D)** Western blots for RSK and S6K phosphorylation upon acute or chronic PDGFRα stimulation. **(E)** Quantification of the Western blotting analyses in (D). **(F)** Human GBMs expressing high levels of PDGF-A have reduced levels of phosphorylation of several members of the translational machinery when compared with GBMs with low levels of PDGF-A. Data represent the mean ± SD of triplicate experiments. The statistical differences were determined using paired $t$ test. *$P < 0.05$ and **$P < 0.01$.
Source data are available for this figure.

suggest that the suppression of translation machinery induced by chronic PDGFRα activation reduces the sensitivity of PDGFRα-positive GBM to the activities of translational inhibitors.

# Discussion

Activation of the PDGF receptors initiates various signaling pathways in many different contexts. Here, we assembled comprehensive maps of all known signaling events emanating from αα and ββ dimer receptors, independent of context, from an extensive review of the literature. The standardized format (SBML and SBGN) used to construct these maps allows updating from community members and makes them amenable to usage on various SBML-compliant tools. These maps represent a new, far-reaching, and extensive resource for anyone studying PDGFR signaling. The maps demonstrate that PDGFR signaling affects most aspects of cellular physiology and that the differences between αα and ββ dimer receptors are readily noticeable. This is supported by observations

in mice that are null for either of the PDGFR genes, which showed profound receptor-specific developmental defects (Soriano, 1994, 1997). This suggests that in mice, the two PDGFRs do not signal identically. Furthermore, seminal in vivo studies interchanging PDGFR intracellular domains revealed a specificity of signaling during development (Klinghoffer et al, 2001), arguing that the signaling arising from PDGFRs is both receptor and context dependent.

The context dependence of signaling is further exemplified by our studies conducted in our PDGFRα-driven GBM mouse model. We developed a new genetically engineered model of GBM based on chronic, autocrine PDGF-A stimulation of overexpressed PDGFRα in the context of loss of p53, genetic events that are clinically supported by the cancer genome atlas data (Cancer Genome Atlas Research Network, 2008; Verhaak et al, 2010; Brennan et al, 2013). Our new model revealed significant aspects of gliomagenesis that were previously unknown. First, we demonstrated an absolute requirement for PDGF-A expression in the context of PDGFRα overexpression for gliomagenesis. This is in line with studies aimed at determining the temporal and sequential genetic events

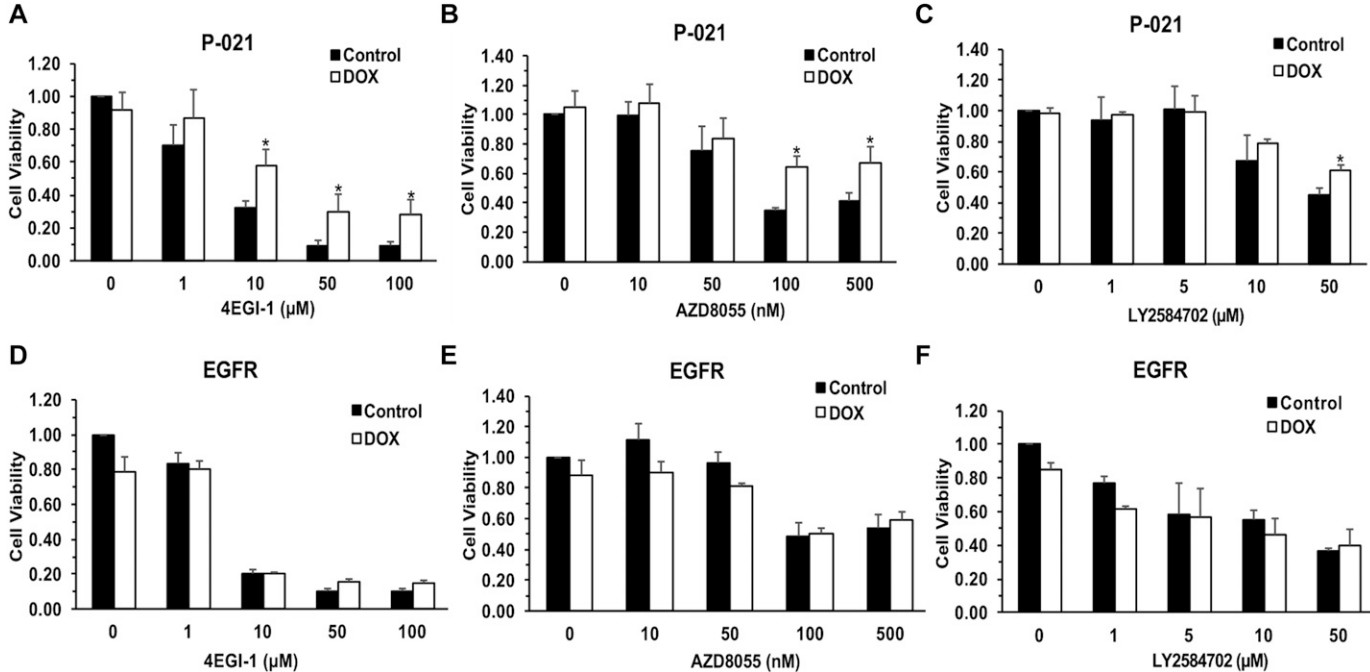

**Figure 7. Chronic PDGFRα activation reduces sensitivity of PDGFRα-positive GBM cells to translational inhibitor–induced suppression on cell viability.**
**(A–C)** Cell viability of PDGFRα GBM primary cell culture with the treatment of 4EGI-1, AZD8055, and LY2584702, respectively, under control or chronic PDGFRα stimulation. **(D–F)** Cell viability of EGFR GBM primary cell culture with the treatment of 4EGI-1, AZD8055, and LY2584702, respectively, under control or chronic PDGFRα stimulation. Cells were treated with the indicated translational inhibitors for 48 h in serum-deficient media, and the cell viability was assessed using the CellTiter-Fluor Cell Viability Assay. Data represent the mean ± SD of triplicate experiments. The statistical differences were determined using paired *t* test. *$P < 0.05$.

in GBM that suggest that overexpression of PDGF-A is an early event in the development of GBM due to low-level broad amplification of chromosome 7 (Ozawa et al, 2014). Second, there is a requisite for loss of tumor suppressor gene function (e.g., p53) for PDGFRα-driven tumorigenesis in mice because overexpression and activation of PDGFRα alone in the absence of p53 loss failed to generate tumor. Our new model offers a unique opportunity to study PDGFRα signaling in a context (glioma)-specific fashion.

Another significant advancement from our work is the analysis of signaling events under acute and chronic receptor activation using state-of-the-art proteomics. Most ex vivo studies of PDGFR signaling relies on acute, saturating receptor stimulation. Although it is a situation that is less representative of clinical settings, it nevertheless established ample information on the varied nature of PDGFR signaling. Here, we demonstrate that the wiring of signaling pathways under chronic PDGFRα activation differs significantly from that of acute PDGFR stimulation, which unveiled a previously unknown function for PDGFR. Stimulation of receptor tyrosine kinases is typically perceived as a positive effect, which translates into a cascade of phosphorylation and other PTM events throughout the cell. A more realistic view, however, is that stimulation of receptors occurs in the context of already established signal wirings, and receptor activation acts positively and negatively upon preexisting feedback and feedforward loops. Therefore, it is not surprising that acute activation of PDGFRα in our GBM cells leads to increase in levels of phosphorylation in 50.3% of detected phosphopeptides and to a decrease in phosphorylation levels in

49.7% of phosphopeptides with varying degrees of amplitude. When comparing with chronic stimulation, however, most of the sites that displayed increases now demonstrate decreases and vice versa, suggesting that prolonged receptor activation dramatically rewires cellular signaling. Acutely stimulated, ligand-mediated dimerization of PDGFRs results in a rapid internalization and degradation of the receptors, which terminates signal outputs. It is possible that chronically activated receptors establish an equilibrium of activation/degradation that is distinct from acute stimulation and conceivably alters ensuing signaling outputs. In addition, constitutively stimulated receptors may also modify their subcellular localization to non–plasma membrane areas where it can be exposed to additional or alternative signaling partners.

By comparing signaling pathways from acute and chronic stimulation of PDGFRα within the same cells, we discovered that chronic PDGFRα signaling attenuates several key components of the translation initiation machinery (schematically represented in Fig S7), which is a new observation for PDGFRα physiology. Much of the protein translational function posited to play a role in cancer has foundation on overexpression and/or enhanced PTM of members of the translation initiation machinery (recently reviewed in Bhat et al [2015]). It is possible that a PDGFRα-mediated reduction in translation initiation will have effects on the translation of specific mRNAs and protein expression. This opens new areas for therapeutic intervention that were otherwise not ascribed to PDGFR signaling where drug sensitivities may occur in a conditional (e.g., PDGFRα-positive cancers) manner.

In conclusion, using comprehensive maps of molecular interactions and signaling events downstream of PDGF receptors, we revealed the context dependency of PDGFRα signaling output in a new model of glioma. We show that chronic activation of PDGFRα leads to a reduction in the activity of a number of translational initiators, and we offer a valuable working model toward a systems-level understanding of the PDGFR network.

# Materials and Methods

### PDGFRα conditional transgenic mice

All mouse procedures were performed in accordance with Beth Israel Deaconess Medical Center's recommendations for the care and use of animals and were maintained and handled under protocols approved by the Institutional Animal Care and Use Committee. Cre/Lox-mediated conditional expression of the human wild-type PDGFRα in the mouse was achieved by generating a targeted knock-in of a cytomegalovirus-actin-globin (CAG)-floxed stop cassette PDGFRα cDNA minigene into the 3′ UTR of the collagen1α1 gene locus as described in the Supplementary Materials and Methods. The LSL-PDGFRα animals were bred to floxed conditional p53 null mice (Marino et al, 2000) to generate compound strains. GBM tumor induction was achieved by stereotactic intracranial injections of adult transgenic mouse (LSL-PDGFRα; p53[2lox]) of a lentivirus that delivers a DOX-inducible expression of human PDGF-A ligand concomitant to a constitutive expression of Cre recombinase as described in detail in the Supplementary Materials and Methods. Details on the cloning, production, and purification of the PDGF-A-Cre lentivirus are described in the Supplementary Materials and Methods. Genotyping of the CAG-LSL-PDGFRα mice is performed by PCR using oligonucleotide primers listed in the Supplementary Materials and Methods from genomic DNA isolated from tail biopsies.

### Mouse GBM primary cultures

Primary cultures of mouse GBM tumors were established as follows: tumors were excised and minced in 0.25% trypsin (wt/vol) and 1 mM EDTA and allowed to disaggregate for 15 min at 37°C. The resulting cell suspension was then strained through a 100-mm cell strainer (Falcon). The single suspension of cells was washed in PBS twice and plated on 0.2% gelatin-coated tissue culture plates. The cells were fed every 24 h with fresh media that consisted of DMEM supplemented with 10% heat-inactivated fetal bovine serum and antibiotics. For DOX treatment, the cells were grown in DMEM 0.1% FBS for 16 h and DOX was added at the indicated final concentrations, and the cells were incubated for the indicated period of time at 37°C in 5% $CO_2$.

### Immunoblots

Western blots were performed as follows: cell lysates were prepared using radioimmunoprecipitation assay buffer supplemented with 5 mM $Na_3VO_4$ (freshly made) and Complete protease inhibitor cocktail (Roche). Equal amount of total cell lysates were subjected to SDS–PAGE and electrotransfered to polyvinylidene difluoride membrane (Immobilon P; Millipore). Blots were blocked in Odyssey blocking buffer (LI-COR) for 1 h on a shaker. Primary antibodies were added to the blocking buffer with 0.2% (vol/vol) TBS Tween 20 (TBS-T) at 1:1,000 dilution and incubated overnight at 4°C on a shaker. The blots were washed several times with TBS-T, and IRDye secondary antibodies (LI-COR) were added at 1:5,000 dilutions into Odyssey blocking buffer with 0.2% (vol/vol) TBS-T and 0.01% SDS and incubated for 1 h at room temperature on a shaker. After several washes, fluorescence signals were detected using Odyssey Clx imaging system as described by the manufacturer (LI-COR). Signal quantification was performed using Image Studio Lite software (LI-COR). The primary antibodies used in these studies are listed in Table S6.

### Proteomic and phospho-proteomic sample preparation

Pooled biological triplicates of flash-frozen cell pellets were lysed with a buffer containing 50 mM Hepes (pH 8.5), 8 M urea, 150 mM NaCl, protease inhibitors (mini-Complete EDTA-free; Roche), and phosphatase inhibitors (PhosSTOP; Roche). The cells were mechanically lysed using a syringe and a 22-gage needle 15 times. Lysates were cleared through centrifugation and protein concentration was determined using a bicinchoninic acid assay (Thermo Fisher Scientific). Equal amounts of protein (4 mg) were reduced for 45 min at 37°C with 5 mM DTT and alkylated with 15 mM IAA for 30 min at 25°C in the dark, before final reduction with 5 mM DTT for 15 min at 25°C. Protein content was then extracted through methanol–chloroform precipitation and resuspended in 50 mM Hepes, 8 M urea, and 150 mM NaCl. For proteolytic digestion, LysC was added at a substrate:enzyme ratio of 100:1 and incubated for 3 h at 37°C. The samples were then diluted to 1.5 mM urea with 50 mM Hepes and digested overnight with trypsin at 25°C with a substrate:enzyme ratio of 50:1. The peptide solutions were then acidified before solid-phase extraction using a Sep-Pak (Waters). Peptide samples were resuspended in 1 ml 50% acetonitrile (ACN) and 2 M lactic acid, and 100 μg of each sample was removed, desalted, and saved for protein-level measurements. Phosphopeptide enrichment was performed as previously described (Erickson et al, 2015).

Non-phosphorylated peptides saved before enrichment and enriched phosphopeptides were then suspended in 100 μl of 200 mM 4-(2-Hydroxyethyl)-1-piperazinepropanesulfonic acid, 4-(2-Hydroxyethyl)piperazine-1-propanesulfonic acid, N-(2-Hydroxyethyl)piperazine-N′-(3-propanesulfonic acid) (pH 8.5) before the addition of 30 μl of anhydrous acetonitrile and 10 μl of a 20 μg/μl stock of tandem mass tag reagent. The samples were incubated for 1 h at 25°C before the addition of 10 μl 5% hydroxylamine. A small portion of each sample was mixed, desalted, and analyzed to determine relative analyte abundance in each sample. The remaining sample was then mixed to ensure equal loading of peptide and phosphopeptide content and acidified before solid-phase extraction using a Sep-Pak. Following isobaric labeling, enriched phosphopeptides were further enriched for phosphotyrosine (pTyr)-containing peptides. Enriched phosphopeptides were resuspended in 450 μl of immunoaffinity purification (IAP) buffer (50 mM MOPS/NaOH, pH 7.2, 10 mM $Na_2PO_4$, and 50 mM NaCl). A phospho-tyrosine–specific antibody (P-Tyr-1000; Cell Signaling Technology) was incubated with protein A agarose beads (Roche) overnight at 4°C in 1% PBS to bind the antibody to the beads. Subsequently, the antibody–bead

mixture was washed 3× with IAP before incubation with enriched phosphopeptides for 1 h at 25°C to enable capture of pTyr-containing peptides. The supernatant, containing enriched phosphopeptides, was removed, de-salted using a Sep-Pak, and saved for offline fractionation. The beads were washed 1× with IAP and 1× with $H_2O$ before performing two elutions using 75 µl of 100 mM formic acid. Enriched pTyr peptides were desalted and resuspended in 1% formic acid before nano liquid chromatography (nLC)–MS/MS analysis. Non-phosphorylated and phosphorylated peptides were fractionated via basic-pH reversed-phase liquid chromatography as previously described (Isasa et al, 2015). Non-phosphorylated samples were resuspended in 5% ACN and 1% formic acid and phosphorylated peptides were resuspended in 1% formic acid before nLC-MS/MS analysis.

## MS and analysis

MS analyses were performed on an Orbitrap Fusion Lumos mass spectrometer (Thermo Fisher Scientific) coupled to an Easy-nLC 1200 ultrahigh-pressure liquid chromatography (LC) pump (Thermo Fisher Scientific). Peptides were separated at 300 nl/min using an analytical column (75-µm inner diameter) that was self-packed with 0.5 cm of Magic C18 resin (5 µm, 100 Å; Michrom Bioresources) followed by 35 cm of Sepax Technologies GP-C18 resin (1.8 µm, 120 Å). LC buffers consisted of 0.1% formic acid (buffer A) and 80% ACN with 0.1% formic acid, and LC gradients were optimized to ensure equal elution of peptides throughout the analysis. Survey scans (MS1) were performed in the Orbitrap (automatic gain control [AGC] target $1 \times 10^6$, 120,000 resolution, and 100 ms maximum injection time) and used to select the 10 most abundant features for MS/MS analysis. Candidate peaks were filtered based on charge sate ≥2 and monoisotopic peak assignment, and dynamic exclusion (60 s ± 10 ppm) was enabled. For non-phosphorylated peptide analysis, only one charge state was selected for each precursor. Precursor ions were isolated (AGC target = $2.5 \times 10^4$) at a width of 0.5 Th using a quadrupole mass filter and fragmented with collision-induced dissociation (35 normalized collision energy) in the ion trap with distinct maximum injection time settings for non-phosphorylated (150 ms) and phosphorylated (200 ms) peptides. To alleviate the effects of precursor ion interference (Ting et al, 2011), multiple fragment ions were isolated (McAlister et al, 2014) using synchronous precursor selection before high energy collisional dissociation (55 NCE, synchronous precursor selection notches = 8, AGC target = $2.2 \times 10^5$, and maximum injection time of 150 or 300 ms for non-phosphorylated and phosphorylated peptides, respectively) MS3 fragmentation and Orbitrap analysis (50,000 resolution).

A compilation of in-house software was used to convert thermo ".raw" mass spectrometric data to mzXML format and to correct monoisotopic $m/z$ measurements and erroneous peptide charge state assignments (Huttlin et al, 2010). The SEQUEST algorithm was used to assign MS/MS spectra to a peptide identification (Eng et al, 1994). Static modifications included tandem mass tag (229.16293 D) on both the N-terminus of peptides and lysine residues and carbamidomethylation of cysteine residues (57.02146 D). Phosphorylation (79.96633 D) was included for phosphopeptide experiments. Peptide spectral matches were filtered to 1% false discovery rate using the target-decoy strategy (Elias & Gygi, 2007),

before being grouped into proteins which were then filtered to 1% false discovery rate at the protein level as previously described (Huttlin et al, 2010). Phosphorylation sites were localized with a modified version of the AScore algorithm, and phosphorylation sites with an AScore > 13 ($P < 0.05$) were considered localized (Beausoleil et al, 2006). Proteins and phosphorylation isoforms were quantified as previously described (Isasa et al, 2015). "Relative abundance" is the expression values for each analyte (protein or phosphorylation isoform) and represent the signal-to-noise value of each sample divided by the sum of all samples for each analyte normalized to 100. For phosphorylated peptides, the quantitative values were normalized to the relative abundance of the protein, to account for changes in protein abundance upon treatment. All data analyses were performed using R (http://www.R-project.org).

## GO enrichment analysis

GO enrichment analysis was performed using the Gene Ontology Consortium, which connects to the analysis tool from the protein annotation through evolutionary relationship (PANTHER) classification system (http://www.pantherdb.org/). The PANTHER system consists of gene function, ontology, pathways, and statistical analysis tools that enable the analysis of large-scale, genome-wide data from sequencing, proteomics, or gene expression experiments (Mi et al, 2013). The GO enrichment analysis was performed using UniProt IDs of the most differentially regulated genes under experimental conditions from our phospho-proteomics data versus a background population of all genes of *Mus musculus* in the PANTHER system.

## Activity flow and process diagrams of PDGFR signaling by CellDesigner

The global activity flow and the comprehensive signaling maps of the PDGFR networks were created with the CellDesigner version 4.4 (http://www.celldesigner.org/), which is a structured diagram editor for drawing gene-regulatory and biochemical networks. The maps are stored using the SBML and are able to link with simulation and other analysis packages through Systems Biology Workbench. All of the species, proteins, reactions, and cellular compartments included in the map are listed in the SBML file when opened by CellDesigner. For individual reactions, PMIDs were added, enabling a direct link to the relevant references.

## Merging of phospho-databases

The merging of the PDGFRα phospho-proteomics data to the PhosphoSitePlus datasets (https://www.phosphosite.org/) (Hornbeck et al, 2015) was performed using an in-house Python script (www.python.org, version 3.6.3) described in details in the Supplementary Materials and Methods.

## Kinase–substrate network visualization using Cytoscape

The kinase substrate phosphorylation network was visualized in Cytoscape, which is a platform for visualizing molecular interaction networks and biological pathways (http://www.cytoscape.org). Molecules in different signaling pathways were assigned with distinct color codes to aid in visualizing the differences. The edges linking kinases

and substrates were colored in accordance with the $\log_2FC$ values of treated versus non-treated in the phospho-proteomics data.

### Cell viability assay

The cell viability was analyzed using the CellTiter-Fluor Cell Viability Assay (Promega), which measures the relative number of viable cells in a population based on the measurement of the conserved and constitutive protease activity within live cells. Briefly, PDGFRα or EGFR cells were seeded in 96-well plates with a density of $0.8 \times 10^4$ cells/well and serum starved overnight. The cells were then treated with 10 μg/ml DOX or vehicle control. The translation inhibitors, including 4EGI-1 (Selleckchem), AZD8055 (Selleckchem), and LY2584702 (Selleckchem), were added at indicated concentrations and incubated for 48 h. The CellTiter-Fluor Reagent was then added to wells and viability measured using a fluorometer ($400\ nm_{Ex}/505\ nm_{Em}$) after incubation for 30 min at 37°C.

### Statistical analyses

Statistical analyses were carried out using GraphPad Prism 7 (GraphPad Software). Two-tailed $t$ tests were used for single comparison. Significance for survival analyses was determined by the log rank (Mantel–Cox) test. $P$-values of less than 0.05 were considered statistically significant.

### Data and materials availability

Proteomics data are available through requests. All materials may be obtained through an material transfer agreement.

# Supplementary Information

# Acknowledgements

We would like to thank Dr. A. Kazlauskas (Schepens Eye Research Institute, Massachusetts Eye and Ear Institute, Boston) for providing the human PDGFRα cDNA, and members of the Charest lab and Dr. Pom P. Ino for critical review of the manuscript. This work was supported by National Institute of Health/National Cancer Institute grants CA185137, CA179563, and CA069246.

## Author Contributions

S Zhou: conceptualization, data curation, formal analysis, validation, investigation, methodology, and writing—original draft, review, and editing.
VA Appleman: investigation and methodology.
CM Rose: resources, data curation, software, investigation, methodology, writing—original draft.
HJ Jun: investigation.
J Yang: data curation and software.
Y Zhou: data curation and software.
RT Bronson: formal analysis and investigation.
SP Gygi: conceptualization, resources, formal analysis, supervision, funding acquisition, investigation, and project administration.
A Charest: conceptualization, resources, data curation, formal analysis, supervision, funding acquisition, validation, investigation, visualization, methodology, writing—original draft, project administration, and writing—review and editing.

## Conflict of Interest Statement

The authors declare that they have no conflict of interest.

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
