## [Reviewer comments · Life Science Alliance]

Chronic platelet derived growth factor receptor signaling exerts control over initiation of protein translation in glioma

Shuang Zhou, Vicky A. Appleman, Christopher M. Rose, Hyun Jung Jun, Juechen Yang, Yue Zhou, Roderick T. Bronson, Steve P. Gygi, and Al Charest DOI: 10.26508/lsa.201800029

Review timeline:

Submission date:	5 February 2018
1 st Editorial Decision:	28 February 2018
Revision received:	9 May 2018
2 nd Editorial Decision:	25 May 2018
2 nd Revision received:	29 May 2018
Accept:	29 May 2018

Report:

(Note: Letters and reports are not edited. The original formatting of letters and referee reports may not be reflected in this compilation.)

1st Editorial Decision

28 February 2018

Thank you for submitting your manuscript entitled "Chronic PDGFR signaling exerts control over initiation of protein translation in glioma" to Life Science Alliance. The manuscript was assessed by expert reviewers, whose comments are appended to this letter. We invite you to submit a revision if you can address the reviewers' key concerns, as outlined here.

As you will see, the referees appreciate your analysis. However, all three consistently point out that much more information need to be included in the manuscript (especially also on the search strategy for the PDGFR signaling map) and that statistical analyses need to be performed. These points seem straightforward to address in a timely manner, and we would thus like to invite your to carefully do so.

Thank you for this interesting contribution to Life Science Alliance. We are looking forward to receiving your revised manuscript.

REFeree REPORTS

Reviewer #1 (Comments to the Authors (Required)):

In this manuscript Zhou and colleagues describe a study of PDGFR signalling, comparing acute versus chronic stimulation and the role of this pathway in glioma. As a starting point the authors curated and defined a map of the PDGF receptor signalling. They then generated a mouse model that can mimic a state of acute PDGFR stimulation by having a cre dependent expression of the receptor and a lentiviral DOX inducible PDGF-A ligand. They established that glioblastoma multiforme (GBM) occurs in TP53 deletion mice in the context of PDGFR constitutive activation. The authors then used a phosphoproteomic approach to compare the protein and phosphoprotein level changes of this model of chronic activation versus an acute and strong activation of the same pathway. They were able to quantify on the order of 7500 proteins and 5700 phosphopeptides. The two different modes of pathway activation resulted in substantial differences in the quantified phosphosites. In order to better understand the differences the authors focused on those phosphosites that have annotated functions and/or kinase regulators. It was unclear to me if the curated map defined at the start of the manuscript was also used in this analysis. Based on a visual inspection of the changed sites with annotation the authors identified "protein translation" as differentially regulated in the two modes of pathway activation. In particular, chronic activation results in the reduction of translation

capacity. This difference was also shown to correlate with a differential effect of translation inhibition by different inhibitors. There are several interesting aspects reported in this study but there several areas where it lacks details. In particular the computational analysis could be strengthened. Overall I think it is a useful contribution to the study of context dependent changes in cell signalling.

Major concerns

1 - While the main findings of this work are interesting and clear the analysis of the data is often confusing. In particular the curation effort at the start of the manuscript appears almost separate from the rest of the work and it is not very well described. More specifically:

1.1 - The manuscript starts by describing what to me sounds like a curation effort to capture the knowledge of the PDGFR pathway in a machine readable format. However the authors provide almost no information about how this was achieved. Was this based on the phosphosite plus database annotations or did the authors curate the 390 article themselves? If the authors curated this themselves, they should provide the reader with a sense of the quality of the curation effort both in terms of accuracy and coverage. Did they establish curation guidelines and was the accuracy accessed by cross curation of some of the same content by independent curators? The authors later mention that most of the phosphosites annotated in this pathway (180 out 189) are found in phosphosite plus sites with a known kinase but they do not mention if the same kinase-site interactions match across the two sets. Also, what is the extent of reactions/interactions in their pathway that are not already covered in similar databases such as reactome or phosphosite plus ?

1.2 - The PDGFR pathway curated by the authors is apparently not used further. As far as I understood from the manuscript the results of the phosphoproteomic experiments are studied using the phosphosite plus annotations for kinase-site interactions and phosphosite functional annotations. What is the purpose of this curation effort then in the context of this study ? Why not use the curated pathway to identify regions in the pathway that are different both in terms of protein abundance across as well as phosphorylation.

2 - There are several remarks in the paper that could be backed up by additional analysis. Concretely:

2.1 - the analysis of the differences in phosphorylation in the two conditions is done by inspecting the differences in the phosphosite changes (Figure 5). There are several standard statistical ways of testing for differences distributions, means or ranks between groups with annotations. It would be straightforward to take the quantified relative changes in phosphosite levels between acute and chronic stimulation and calculate a p-value for differential regulation for each of the annotations. This can be done for all phosphosites belonging to a functional group (e.g. RNA binding and translation) and kinase (e.g. all AKT1 targets). This can be done using a rank test (Kolmogorov-Smirnov test) or permutation based test (Gene Set Enrichment Analysis). Both of these tests will take into account the log(FC) without having to define a specific cut-off value. This will give the authors a ranked list kinases and functional groups that have the most significant changes in regulation.

2.2 - The authors claim that: "The overall effect of chronic stimulation of PDGFR α is a significant reversal in the phospho-events when compared to acute stimulation". This would suggest that the changes in both are anti-correlated but this is never shown or tested. Are the log₂ FC(chronic/cont) anti-correlated with the log₂ FC(acute/cont) ? The representation of the data in figure 4 is not intuitive. Why not plot directly the log₂ FC(chronic/cont) vs log₂ FC(acute/cont) ? Based on the plot in figure 4 it may be that most of the changes occur only in the acute signalling and that the chronic condition shows almost no change in phosphorylation at steady state. If this is the case, it is an important point to make. One would expect that the chronic stimulation would result in protein abundance differences primarily. The authors say: "3.5% of the phospho-peptides showed significant variations between chronic and acute PDGF-A stimulation" - how was significance measured ?

3 - In regards to the technical quality of proteomics measurements, what are the correlations of the biological replicates for the protein measurements and phosphoproteomic measurements ?

4 - The proteomics measurements obtained in the study needs to available in supplementary results

Minor concerns

- The description of the curated PDGFR pathway models can be summarized much more succinctly and more emphasis given to the quality of the curation (accuracy and coverage). There is very little in this section that influences the rest of the manuscript and a very long list elements that could be a table in supplementary results.

- It is not clear if the phosphoproteomic measurements are also in triplicate. How were the normalized value of phosphorylation obtained in the context of the replicates ?

- The description of the analysis of the annotated phosphosites from phosphosite plus needs to be improved (see major concerns) and can also be summarized. The first paragraph in particular of the section "Chronic PDGFR α activation..." can be summarized further. Mapping phosphorylation positions between phosphosite plus and the author's dataset should not be described as a "method" nor a "decision algorithm". In addition, it was not clear if the authors used the information in phosphosite plus from human proteins or just the mouse proteins. If the human information was used, how were the orthology assignments made ?

- I found it interesting that the authors used the RPPA data from TCGA to test for differences in phosphosites for low versus high PDGF-a GBM tumours. It goes beyond the scope of the work but there is also RPPA data for cancer cell lines that have been tested for survival under a large panel of drugs (<https://doi.org/10.1016/j.ccell.2017.01.005>). The authors may consider also testing for differences in the effects of translation inhibitors according to differences in PDGF-A or downstream activation of kinases.

- As described in the discussion section, the chronic activation of the pathway will result in differences in protein abundances level that likely desensitize the pathway. The authors could have tried to use the protein abundance level data to study this.

- Figure 2 is not very useful for a reader - Not much can be see from it at this level of resolution in a manuscript.

Reviewer #2 (Comments to the Authors (Required)):

Chronic platelet derived growth factor receptor signaling exerts control over initiation of protein translation in glioma

The manuscript is constructed around four topics: 1) a comprehensive PDGFR signaling map based on literature mining. 2) generation of a genetically engineered mouse model (GEMM) that recapitulates human PDGFR signaling in glioma (GBM). 3) an elegant isobaric labeling proteomics and phosphoproteomics experiment comparing acute with chronic PDGFR signaling in GBM cell lines and a tailored bioinformatics workflow to extract biological insight from the proteomics data in the context of existing phosphositePlus kinase/phosphosite relationships. 4) down regulation of translation initiation in chronic PDGFR signaling in GBM.

The manuscript is well written and clear. Experiments are well designed and technically carried out very well. Appropriate controls are included. Cell line Proteomics findings are backed-up by western-blot validation in GEMM tissue samples. The PDGFR signaling GEMM constitutes a useful research resource for glioma research, which has traditionally been focused on the more prevalent EGFRvIII mutant driven glioma.

However, several topics raise some concern:

First: although the PDGFR signaling map is annotated with PubMed entries underpinning the association of proteins with PDGFR signaling, no description of the actual search strategy or criteria is described the (supplementary) methods section. This should be included. The last paragraph of the result section on "activity flow and process diagrams of PDGFR signaling" is too much an iteration of numbers of components and compartments. A shorter description in the main manuscript and a comprehensive description of components and compartments in supplementary materials improves readability. The effort of constructing the signaling map in Celldesigner in SBML and SBGN compliant format enables reuse of the map by the research community.

Second: the authors focus on phosphosite quantitative values by normalizing for protein expression changes. However in the cells as well as in the tumor overall signal transduction is a function of both the phosphorylation level of proteins (and sites) as well as corresponding protein abundance. This is especially important in an experiment comparing the effect of short time stimulation (15 min) vs. chronic stimulation (48 hrs.). In the chronic stimulation population protein synthesis will occur (as well as in the corresponding control) and might be stimulus-specific (which is not to be expected for the control). The authors should reanalyze the phosphoproteomics data without normalization to uncover the effect -if any- of protein abundance during acute and chronic PDGFR stimulation.

Third: chronic PDGFR signaling is the clinically relevant condition in 13% of GBM cases. The authors show that biological processes associated with differential signaling in chronic PDGFR signaling points at lowered signaling for "RNA binding and translation related" processes, and the functional analysis and therapeutic potential of this process constitutes the rest of the discussion. However, looking at fig 5A, other important processes such as "Transcription and cell cycle", "MAPK signaling" and "STAT & Receptors" show equally lowered signaling in the chronic case. The authors should explain either why they only focused on translation initiation or also include a more detailed analysis of other important down regulated processes. Also the upregulation of the Cadherin binding GO term (fig 4C) deserves exploration.

Fourth: in the current manuscript, the main finding of the phosphoproteomics experiment is that translation initiation associated phosphosites show lower phosphorylation in chronic PDGFR signaling. This is the clinically relevant case in GBM (fig 6). Subsequently, the author show that translation initiation inhibitors do not really affect cell viability in chronic PDGFR signaling. Although the experiment is sound and well conducted (fig 7), I doubt the (clinical) relevance. Since the translation initiation machinery is not very active, inhibition will most likely not add anything in reducing cell viability, and this is shown. I'm not sure this is the same as conferring resistance to translational inhibitors (title fig 7). In cancers where translation initiation is UP (NSCLC, prostate cancer) these inhibitors do show potential.

Reviewer #3 (Comments to the Authors (Required)):

In the present manuscript, the authors develop a bioinformatics approach to extract relevant information from phosphoproteomics dataset. They applied this workflow to characterize signaling events triggered by chronic and acute stimulation of the PDGF receptor alpha in a mouse model of glioma. While the method the authors developed do not represent a real novelty in the field (see S. Munk, *Methods Mol Biol* 2016; C. D. Terfve, *Nat Commun* 2015; F. Sacco, *Proteomics* 2017, S. J. Humphrey et al., *Nat Biotech* 2015), the biological question and the generated mouse model could be of general interest for the scientific community. Finally, the authors found that while transient stimulation activates several key components of the translation initiation machinery (this was also already known), the clinically relevant chronic activity of PDGFR α is associated with a significant shut down of translational members.

In my opinion, the paper could be of general interest, but a number of major issues need to be addressed:

-In the first part of the manuscript, the authors made a great effort to manually assemble a PDGF signaling network. How the 390 publications used to construct this network were selected? Did the authors employ text-mining approaches?

-Regarding the MS-based phosphoproteomic approach, what is the correlation between biological replicates? In each condition and replicate, the number of identified phosphopeptides was comparable? Since the authors mentioned also the proteomic data, can they add more technical info about the proteome?

-It is not clear which statistical methods the authors applied to identify the acutely and chronically modulated phosphosites by PDGF. Additionally, what is the number of statistically significant phosphosites modulated by acute and chronic PDGF stimulation? Although chronic and acute stimulation are inversely correlated, are there some phosphosites that are regulated by both the type of stimulations?

-For the GO enrichment analysis, what is the FDR threshold that the authors used (Fig. 4C)?

-The phosphoproteomics approach was able to recapitulate some of the known PDGFR substrates?

The authors should also compare their phosphoproteomic dataset with the one already published.

-In the chronic PDGF stimulation, the authors induce the PDGFR expression by treating cells with doxocycline for 48h. After 48h, it is very likely that protein expression is also modulated. The authors stated that they also quantified the proteome by a MS-based approach, however they did not comment it. Some of the changes observed at the phosphorylation level in the chronically stimulated cells might also be due to a different expression of the proteins. Did the authors check it?

This is an important issue: as the authors hypothesized, chronic and acute stimulation might differ because signaling network are rewired, implicating that some connections might be lost because of the modulation of the concentration of key signaling proteins.

-To interpret phosphoproteomic dataset and get some insights about PDGFR-dependent signaling, the authors mapped their dataset onto the kinase-substrate network extracted from Phosphositeplus. Figure 5 is the main result of the manuscript, being the proof that the developed bioinformatics workflow can successfully be used to extract meaningful information from phosphoproteomic dataset. However, the two generated networks are not informative at all: it is not clear whether the modulation of phosphorylation also mirrors a regulation of protein activity; the phosphorylation and activation of PDGFR downstream signaling proteins is also unknown. This is somehow the major issue of the paper.

-Recently many approaches that combine phosphoproteomics data with kinase-substrates network have been developed and recently reviewed. The authors should cite at least some of them (S. Munk, *Methods Mol Biol* 2016; C. D. Terfve, *Nat Commun* 2015; F. Sacco, *Proteomics* 2017).

1st Revision – authors' response

9 May 2018

Reviewer #1 (Comments to the Authors (Required)):

Major concerns

1 - While the main findings of this work are interesting and clear the analysis of the data is often confusing. In particular the curation effort at the start of the manuscript appears almost separate from the rest of the work and it is not very well described. More specifically:

1.1 - The manuscript starts by describing what to me sounds like a curation effort to capture the knowledge of the PDGFR pathway in a machine readable format. However the authors provide almost no information about how this was achieved. Was this based on the phosphosite plus database annotations or did the authors curate the 390 article themselves? If the authors curated this themselves, they should provide the reader with a sense of the quality of the curation effort both in terms of accuracy and coverage. Did they establish curation guidelines and was the accuracy accessed by cross curation of some of the same content by independent curators?

First and foremost, we appreciate reviewer #1's diligence and attention to details. To address this question, we have added the requested information for the curation effort into the supplementary materials under the section "**Curation criteria for the PDGFR signaling map**". Briefly, the knowledge gathered to build the PDGFR pathway was curated by ourselves through several steps, which included keyword searches, and well defined criteria for inclusion or exclusion of manuscripts and their contents. Several iterations were performed with new keywords and the preliminary map was cross-referenced to similar maps for EGFR and mTOR (Oda et al., *Mol Syst Biol*, 2005, Caron et al., *Mol Syst Biol*, 2010) for accuracy and consistency. We have added text to the main manuscript directing readers to the Supplementary Information section for more details on the curation process.

The authors later mention that most of the phosphosites annotated in this pathway (180 out 189) are found in phosphosite plus sites with a known kinase but they do not mention if the same kinase-site interactions match across the two sets. Also, what is the extent of reactions/interactions in their pathway that are not already covered in similar databases such as reactome or phosphosite plus?

We would like to thank the reviewer for this thoughtful comment and we agree that the two parts of the manuscript need to be linked. We have summarized this information into a new Table S2 and

refer to this new table in the main text of the manuscript. Briefly, the table lists the number of events (Reactions, State Transitions and Phosphorylation sites) from our map that are covered (or not) in the Reactome and PhosphoSite plus databases.

1.2 - The PDGFR pathway curated by the authors is apparently not used further. As far as I understood from the manuscript the results of the phosphoproteomic experiments are studied using the phosphosite plus annotations for kinase-site interactions and phosphosite functional annotations. What is the purpose of this curation effort then in the context of this study? Why not use the curated pathway to identify regions in the pathway that are different both in terms of protein abundance across as well as phosphorylation.

We would like to thank the reviewer for this constructive comment. The purpose of the curation effort for the PDGFR pathway is to 1) establish a comprehensive map for PDGFR signaling which is currently lacking (current visualizations are often over simplistic). The effort of constructing the signaling map in CellDesigner in SBML compliant format enables reuse of the map by the research community. Since the map is constructed in a non-restrictive manner, ie. from any disease, tissue or species, it is meant to be as comprehensive as possible. It is easy to imagine then that not all members of a given phosphoproteomics dataset derived from a specific disease/tissue will be represented in the map and vice versa. 2) To assist in our understanding of the phosphoproteomics data and the PhosphoSite Plus database. As you suggested, the PDGFR map has been used to i) identify regions in the pathway that are differentially regulated by acute vs chronic PDGFR stimulation (Figure 5) from the phosphoproteomics data; ii) match with the PhosphoSite Plus and Reactome database and compare the coverage on phosphorylation sites and reactions (Table S2).

2 - There are several remarks in the paper that could be backed up by additional analysis. Concretely:

2.1 - the analysis of the differences in phosphorylation in the two conditions is done by inspecting the differences in the phosphosite changes (Figure 5). There are several standard statistical ways of testing for differences distributions, means or ranks between groups with annotations. It would be straightforward to take the quantified relative changes in phosphosite levels between acute and chronic stimulation and calculate a p-value for differential regulation for each of the annotations. This can be done for all phosphosites belonging to a functional group (e.g. RNA binding and translation) and kinase (e.g. all AKT1 targets). This can be done using a rank test (Kolmogorov-Smirnov test) or permutation based test (Gene Set Enrichment Analysis). Both of these tests will take into account the log(FC) without having to define a specific cut-off value. This will give the authors a ranked list kinases and functional groups that have the most significant changes in regulation.

As suggested by the reviewer, we have performed a Kolmogorov-Smirnov test statistical analysis. The ranked lists of functional groups (Table S4) and kinases (Table S5) have been included in the updated manuscript. Among all the functional groups, RNA binding and translation pathway showed the most significantly different distribution under acute vs. chronic PDGFR stimulation ($P < 0.05$). The Reference Figure 1 below is the representative figure for the Kolmogorov-Smirnov test for RNA binding and translation pathway demonstrating statistical significance. The most differentially influenced kinases include mTOR, p90RSK, ERK2, and JNK3 (Table S5).

Reference Figure 1

Reference Figure 1. distribution of the Log2 for p-sites in RNA translation pathway chronic and acute of PDGFR α . Two Kolmogorov-Smirnov performed ($P < 0.05$,

Cumulative FC values binding and under stimulation sample test was $D = 0.2361$).

2.2 - The authors claim overall effect of chronic of PDGFR α is a reversal in the phospho-compared to acute

that: "The stimulation significant events when stimulation".

This would suggest that the changes in both are anti-correlated but this is never shown or tested. Are the $\log_2 FC(\text{chronic}/\text{cont})$ anti-correlated with the $\log_2 FC(\text{acute}/\text{cont})$? The representation of the data in figure 4 is not intuitive. Why not plot directly the $\log_2 FC(\text{chronic}/\text{cont})$ vs $\log_2 FC(\text{acute}/\text{cont})$? Based on the plot in figure 4 it may be that most of the changes occur only in the acute signalling and that the chronic condition shows almost no change in phosphorylation at steady state. If this is the case, it is an important point to make. One would expect that the chronic stimulation would result in protein abundance differences primarily. The authors say: "3.5% of the phospho-peptides showed significant variations between chronic and acute PDGF-A stimulation" - how was significance measured?

We appreciate the reviewer's comment and would like to point out that the chronic condition shows considerable changes in phosphorylation. As shown in the Reference Figure 2 below (a direct plot for the $\log_2 FC(\text{chronic}/\text{cont})$ vs $\log_2 FC(\text{acute}/\text{cont})$), the majority of the phospho-sites showed changes upon chronic or acute PDGF-A stimulation, while only a small part of the phospho-sites showed reversal effects (left up or right bottom). We maintain that Figure 4C enables a more apparent visualization of the changes upon chronic and acute PDGF-A stimulation. The 3.5% phospho-peptides were the outlier phosphorylation events with a cut-off value of $\log_2 FC < -1$ or $> +1$. We have changed the text in the main manuscript to avoid confusion.

Reference Figure 2

Reference Figure 2. Plot of differential Log2 FC of phosphorylated peptides upon acute or chronic stimulation of PDGFR α .

3 - In regards to the technical quality of proteomics measurements, what are the correlations of the biological replicates for the protein measurements and phosphoproteomic measurements?

In this study, we used pooled biological triplicates for the total protein and phosphoproteomic MS/MS analyses.

4 - The proteomics measurements obtained in the study needs to available in supplementary results

Thank you for pointing out this omission. The proteomics data have been added into the supplementary documents as Data file 8.

Minor concerns

- The description of the curated PDGFR pathway models can be summarized much more succinctly and more emphasis given to the quality of the curation (accuracy and coverage). There is very little in this section that influences the rest of the manuscript and a very long list elements that could be a table in supplementary results.

Thank you for your comment. Changes have been made according to your suggestion. A table of elements has been added in Supplementary Information as Table S1.

- It is not clear if the phosphoproteomic measurements are also in triplicate. How were the normalized value of phosphorylation obtained in the context of the replicates?

The measurements were from pooled triplicates.

- The description of the analysis of the annotated phosphosites from phosphosite plus needs to be improved (see major concerns) and can also be summarized. The first paragraph in particular of the section "Chronic PDGFR α activation..." can be summarized further. Mapping phosphorylation positions between phosphosite plus and the author's dataset should not be described as a "method" nor a "decision algorithm". In addition, it was not clear if the authors used the information in phosphosite plus from human proteins or just the mouse proteins. If the human information was used, how were the orthology assignments made ?

We have compressed the first paragraph of the section “Chronic PDGFR α activation orchestrates repression of protein translation initiation” as suggested and eliminated the language regarding “decision algorithm”. For the PhosphoSite Plus data, we used phosphorylation information from all the species, including human, mouse, rat, cow, chicken, and rabbit.

- I found it interesting that the authors used the RPPA data from TCGA to test for differences in phosphosites for low versus high PDGF- α GBM tumours. It goes beyond the scope of the work but there is also RPPA data for cancer cell lines that have been tested for survival under a large panel of drugs (<https://doi.org/10.1016/j.ccell.2017.01.005>). The authors may consider also testing for differences in the effects of translation inhibitors according to differences in PDGF-A or downstream activation of kinases.

Thank you very much for this insightful comment. This RPPA dataset generated by Li et al. (2017) did not assay the levels of PDGF-A or the phosphorylation of PDGFR α . However, based on these RPPA data for cancer cell lines, we were able to test the correlation for the drug sensitivity of translation inhibitors with the activation of translation related proteins. As shown in the volcano plots below (Reference Figure 3), for all of the 4 tested inhibitors against RNA translation pathway in MCLP (MD Anderson Cell Lines Project <http://tcpaportal.org/mclp/#/>), the phosphorylation of 4EBP1 was correlated (though not all were statistically significant) with a negative impact on the drug IC50 value. Similar effects for phospho-4EBP1 and phospho-S6 were also observed for the mTOR inhibitors (Data not shown). These data suggest a possible positive correlation for the phosphorylation of RNA translation related proteins with translation inhibitor drug sensitivity, which is consistent with our finding in this manuscript.

Reference Figure 3:

Reference Figure 3. The correlation between sensitivity of the indicated translation inhibitors with the phosphorylation of translation related protein. As shown by the Spearman's Rho, the phosphorylation of 4EBP1 was correlated with a negative impact (indicated by arrows) on the indicated drug IC50 values. The RPPA and drug sensitivity data for cancer cells were curated from MCLP (MD Anderson Cell Lines Project <http://tcpportal.org/mclp/#/>). YK-4-279 (RNA Helicase A inhibitor); Omacetaxine mepesuccinate (ribosome inhibitor); SR-II-138A (silvestrol analog, translation inhibitor); CR-1-31B (eIF4A RNA helicase inhibitor).

- As described in the discussion section, the chronic activation of the pathway will result in differences in protein abundances level that likely desensitize the pathway. The authors could have tried to use the protein abundance level data to study this.

This is another insightful comment from the reviewer. We have used the protein abundance level data and the phosphoproteomics data without normalization to study this question. Our data show that 1) except for a few outliers, the differences in protein abundance levels between chronic or acute activation of PDGFRa pathway is not very dramatic (Reference Figure 4) and 2) performing a similar analysis as in Figure 4C with non-normalized phospho-data the functionalities (Go analysis) for the differentially regulated phosphorylation sites, from outlier phosphorylation events (Log₂FC <-1, >+1) revealed similar results (RNA binding and translation pathway) ranking the top categories (Reference Figure 5).

Reference Figure 4:

Reference Figure 4. Analysis of differential Log2 FC on protein abundance levels under acutely and chronically stimulated PDGFR α signaling. The majority of the proteins show minimal change under the two conditions. Proteolipid Protein 2 (PLP2), which was most differentially regulated (Log2FC <-1, >+1), is indicated in the plot.

Reference Figure 5:

Reference Figure 5. Phospho-proteomics analysis of acutely and chronically stimulated PDGFR α signaling without normalization to protein abundance. On the left and right top are the potential most differentially up (left) and down (right) regulated signaling functionalities upon chronic vs. acute stimulation of PDGFR α identified by GO term enrichment analysis. Only the categories with p-value smaller than 0.05 are displayed. False Discovery Rate (FDR) < 0.05.

- Figure 2 is not very useful for a reader - Not much can be seen from it at this level of resolution in a manuscript.

Figure 2 is present to show a general view of the PDGFR signaling and the functional groups. A high resolution PDF version of Figure 2 and the SBML file of Figure 2 are included in the Supplementary Information section as Data Files 1 and 3.

Reviewer #2 (Comments to the Authors (Required)):

However, several topics raise some concern:

First: although the PDGFR signaling map is annotated with PubMed entries underpinning the association of proteins with PDGFR signaling, no description of the actual search strategy or criteria is described in the (supplementary) methods section. This should be included. The last paragraph of the result section on "activity flow and process diagrams of PDGFR signaling" is too much an iteration of numbers of components and compartments. A shorter description in the main manuscript and a comprehensive description of components and compartments in supplementary materials improves readability. The effort of constructing the signaling map in CellDesigner in SBML and SBGN compliant format enables reuse of the map by the research community.

We would like to thank the reviewer for his/her constructive comments and have made changes accordingly. As suggested by reviewer #1 also, we have added a description of the actual curation strategy into the Supplementary Information section. We have also eliminated the last paragraph of the "activity flow and process diagrams of PDGFR signaling" section and replaced it with Table S1, which lists the components and compartments of the PDGFR signaling.

Second: the authors focus on phosphosite quantitative values by normalizing for protein expression changes. However in the cells as well as in the tumor overall signal transduction is a function of both the phosphorylation level of proteins (and sites) as well as corresponding protein abundance. This is especially important in an experiment comparing the effect of short time stimulation (15 min) vs. chronic stimulation (48 hrs.). In the chronic stimulation population protein synthesis will occur (as well as in the corresponding control) and might be stimulus-specific (which is not to be expected for the control). The authors should reanalyze the phosphoproteomics data without normalization to uncover the effect -if any- of protein abundance during acute and chronic PDGFR stimulation.

This concern was also raised by reviewer #1. We analyzed protein levels separately from the phosphoproteomics data without normalization. As shown in the Reference Figures 4 and 5 above, protein levels do not change dramatically between acute and chronically stimulated PDGFRa. In addition, analyzing the non-normalized phosphoproteomics data similarly to Fig. 4C resulted in similar outcome.

Third: chronic PDGFR signaling is the clinically relevant condition in 13% of GBM cases. The authors show that biological processes associated with differential signaling in chronic PDGFR signaling points at lowered signaling for "RNA binding and translation related" processes, and the functional analysis and therapeutic potential of this process constitutes the rest of the discussion. However, looking at fig 5A, other important processes such as "Transcription and cell cycle", "MAPK signaling" and "STAT & Receptors" show equally lowered signaling in the chronic case. The authors should explain either why they only focused on translation initiation or also include a more detailed analysis of other important down regulated processes. Also the upregulation of the Cadherin binding GO term (fig 4C) deserves exploration.

To address this comment, we have performed a statistical analysis (Kolmogorov-Smirnov test) on the phosphopeptides with kinase-substrate information. We summarized the ranked lists of functional groups (Table S4) and kinases (Table S5) and among all the functional groups, Cap dependent RNA translation pathway showed the most significantly different distribution under acute vs. chronic PDGFRa stimulation. This analysis helped us decide to focus on translation down regulation.

Fourth: in the current manuscript, the main finding of the phosphoproteomics experiment is that translation initiation associated phosphosites show lower phosphorylation in chronic PDGFR signaling. This is the clinically relevant case in GBM (fig 6). Subsequently, the author show that translation initiation inhibitors do not really affect cell viability in chronic PDGFR signaling. Although the experiment is sound and well conducted (fig 7), I doubt the (clinical) relevance. Since the translation initiation machinery is not very active, inhibition will most likely not add anything in reducing cell viability, and this is shown. I'm not sure this is the same as conferring resistance to

translational inhibitors (title fig 7). In cancers where translation initiation is UP (NSCLC, prostate cancer) these inhibitors do show potential.

We appreciate the reviewer's keen sense of observation in this comment. Yes, we agree with the reviewer that our claim that PDGFR α expression will confer resistance to inhibitors of translation is somewhat overstated and as such we toned down the text describing this work in the manuscript. We would like to point out to the reviewer our analysis of RPPA data for cancer cell lines and their sensitivity to translation inhibitors that shows that phosphorylation of 4EBP1 was correlated with a negative impact on the drugs IC50 values (Reference Figure 3 above). This data suggest a possible positive correlation for the phosphorylation of RNA translation related proteins with translation inhibitor drug sensitivity, which is consistent with our finding in this manuscript. This requires additional future investigations to better understand this relationship mechanistically. We have also changed the title of Fig 7 to be more accurate.

Reviewer #3 (Comments to the Authors (Required)):

In my opinion, the paper could be of general interest, but a number of major issues need to be addressed:

-In the first part of the manuscript, the authors made a great effort to manually assemble a PDGF signaling network. How the 390 publications used to construct this network were selected? Did the authors employ text-mining approaches?

We have addressed the curation strategy above and included a detailed description into the Supplementary Information section.

-Regarding the MS-based phosphoproteomic approach, what is the correlation between biological replicates? In each condition and replicate, the number of identified phosphopeptides was comparable? Since the authors mentioned also the proteomic data, can they add more technical info about the proteome?

As mentioned already, the phosphoproteomics was performed using pooled biological triplicates. The technical information about the proteome has been added into the "Proteomic and phosphoproteomic sample preparation" section in Materials and Methods.

-It is not clear which statistical methods the authors applied to identify the acutely and chronically modulated phosphosites by PDGF. Additionally, what is the number of statistically significant phosphosites modulated by acute and chronic PDGF stimulation? Although chronic and acute stimulation are inversely correlated, are there some phosphosites that are regulated by both the type of stimulations?

We would like to thank the reviewer for this constructive comment. As shown in Figure 4C, we compared acute to chronic PDGFR α stimulation by analyzing the differential between chronically stimulated and acutely stimulated samples (Fig. 4C, y axis). The majority (96.5%) of the phosphopeptides display minimal changes (Log2FC between -1 to 1) between control, acute and chronic ligand stimulation, however, 3.5% of the phospho-peptides showed variations $>+1$ and <-1 Log2FC between chronic and acute PDGF-A stimulation (Fig. 4C). The majority of the phosphosites were regulated similarly by both types of stimulations, while only a small part of the phosphosites showed reversal effects. We have updated our statements in the manuscript to avoid any confusion.

-For the GO enrichment analysis, what is the FDR threshold that the authors used (Fig. 4C)?

FDR <0.05 was used as the FDR threshold. This information has been added into the legend of Figure 4.

-The phosphoproteomics approach was able to recapitulate some of the known PDGFR substrates? The authors should also compare their phosphoproteomic dataset with the one already published.

As already mentioned above, we compared our phosphoproteomics with the Phosphorylation_Site dataset of the Phosphosite Plus database. Our phosphoproteomics was able to recapitulate some of the known PDGFR substrates, and as shown in Figure S5, most phosphosites in our PDGFR phosphoproteomics were observed in the Phosphorylation_Site dataset, except for 558 sites that are considered newly identified.

-In the chronic PDGF stimulation, the authors induce the PDGFR expression by treating cells with doxocycline for 48h. After 48h, it is very likely that protein expression is also modulated. The authors stated that they also quantified the proteome by a MS-based approach, however they did not comment it. Some of the changes observed at the phosphorylation level in the chronically stimulated cells might also be due to a different expression of the proteins. Did the authors check it?

This is an important issue: as the authors hypothesized, chronic and acute stimulation might differ because signaling network are rewired, implicating that some connections might be lost because of the modulation of the concentration of key signaling proteins.

This comment has been addressed above. Briefly, as you suggested, we analyzed the phosphoproteomics data without normalization and we analyzed the proteomics data for changes in protein levels between chronic vs acute stimulation. As shown in the Reference Figures 4 and 5 above, when analyzing the functionalities for the differentially regulated phosphorylation sites, we obtained similar results as with the normalized data, that is RNA binding pathway was among the top ranking categories (Reference Figure 5). Analyzing total proteomics data between chronic and acute stimulation revealed that there are little differences in protein levels (Reference Figure 4).

-To interpret phosphoproteomic dataset and get some insights about PDGFR-dependent signaling, the authors mapped their dataset onto the kinase-substrate network extracted from Phosphositeplus. Figure 5 is the main result of the manuscript, being the proof that the developed bioinformatics workflow can successfully be used to extract meaningful information from phosphoproteomic dataset. However, the two generated networks are not informative at all: it is not clear whether the modulation of phosphorylation also mirrors a regulation of protein activity; the phosphorylation and activation of PDGFR downstream signaling proteins is also unknown. This is somehow the major issue of the paper.

This comment was also raised by the other reviewers and has been addressed above.

-Recently many approaches that combine phosphoproteomics data with kinase-substrates network have been developed and recently reviewed. The authors should cite at least some of them (S. Munk, Methods Mol Biol 2016; C. D. Terfve, Nat Commun 2015; F. Sacco, Proteomics 2017).

We would like to thank the reviewer very much for suggesting the addition of these original manuscripts and bringing this to our attention. We have cited these papers in our updated manuscript.

Thank you for submitting your revised manuscript entitled "Chronic PDGFR signaling exerts control over initiation of protein translation in glioma". We would be happy to publish your paper in Life Science Alliance pending that you provide the source data for all western blots and a revised manuscript to address the two remaining concerns of reviewer #1.

REFEREE REPORTS

Reviewer #1 (Comments to the Authors (Required)):

The authors have addressed most of my previous concerns. They have simplified and reduced several parts that made the article harder to follow and have added statistical testing for some of the claims.

There are only 2 small points that the authors did not address that I still think need to be changed:
1 - Regarding figure 4C, the authors claim that there is a reversal of the phosphorylation events between chronic and acute. They use the fit of the regression to and r^2 to support this claim. This is incorrect and will mislead several readers. As the authors show in the reply to my concerns (Reference Figure 2 of the report) the acute and chronic response are generally positively correlated ($r^2=0.22$). The plot the authors choose for Fig 4A is used to highlight the differences. I accept that the authors prefer to use this plot to highlight the differences but fitting a regression to this curve to claim a strong reversal is highly misleading. The values for $\text{Log}_2 \text{FC}(\text{Acute}/\text{control})$ are both in the x axis and y axis (subtracted from the $\text{Log}_2 \text{FC}(\text{Chronic}/\text{control})$). Of course there will be such a strong linear dependency in such a graph. If $\text{Log}_2 \text{FC}(\text{Chronic}/\text{control})$ was a random variable you would still see such a strong linear anti-correlation. The authors should remove the linear fit from the text and limit the claim for the reversal.

2 - In reply to my concerns the authors claimed to have added the proteomics data as an additional supplementary dataset but I could not find it. This data needs to be available as part of the manuscript as a spreadsheet or text file with the phosphorylation and protein abundance values measured for chronic and acute stimulations.

Reviewer #3 (Comments to the Authors (Required)):

The authors carefully revised the presented manuscript, which is now drastically improved.

2nd Revision – authors' response

29 May 2018

Reviewer #1 (Comments to the Authors (Required)):

The authors have addressed most of my previous concerns. They have simplified and reduced several parts that made the article harder to follow and have added statistical testing for some of the claims.

There are only 2 small points that the authors did not address that I still think need to be changed:
1 - Regarding figure 4C, the authors claim that there is a reversal of the phosphorylation events between chronic and acute. They use the fit of the regression to and r^2 to support this claim. This is incorrect and will mislead several readers. As the authors show in the reply to my concerns (Reference Figure 2 of the report) the acute and chronic response are generally positively correlated ($r^2=0.22$). The plot the authors choose for Fig 4A is used to highlight the differences. I accept that the authors prefer to use this plot to highlight the differences but fitting a regression to this curve to claim a strong reversal is highly misleading. The values for $\text{Log}_2 \text{FC}(\text{Acute}/\text{control})$ are both in the x axis and y axis (subtracted from the $\text{Log}_2 \text{FC}(\text{Chronic}/\text{control})$). Of course there will be such a strong linear dependency in such a graph. If $\text{Log}_2 \text{FC}(\text{Chronic}/\text{control})$ was a random variable you would still see such a strong linear anti-correlation. The authors should remove the linear fit from the text and limit the claim for the reversal.

We would like to thank the Reviewer for his/her thoughtful request and we have made changes accordingly. The linear fit line has been removed from Figure 4C and we have also limited the claim for the reversal. The sentence “The overall effect of differential the phospho-events when compared to acute stimulation (negative slope: $y=-0.69x+0.01$ $r^2=0.58$).” has been removed from the text.

2 - In reply to my concerns the authors claimed to have added the proteomics data as an additional supplementary dataset but I could not find it. This data needs to be available as part of the manuscript as a spreadsheet or text file with the phosphorylation and protein abundance values measured for chronic and acute stimulations.

We apologize and we thank you for pointing this out. The source data has been uploaded.

Reviewer #3 (Comments to the Authors (Required)):

The authors carefully revised the presented manuscript, which is now drastically improved.

between chronic stimulation and acute stimulation of PDGFR is an important reversal in

Thank you very much for your positive comments and your help on improving our manuscript.